# Phenothiazines Inhibit SARS-CoV-2 Entry through Targeting Spike Protein

**DOI:** 10.3390/v15081666

**Published:** 2023-07-31

**Authors:** Taizhen Liang, Shiqi Xiao, Ziyao Wu, Xi Lv, Sen Liu, Meilin Hu, Guojie Li, Peiwen Li, Xiancai Ma

**Affiliations:** 1Guangzhou National Laboratory, Guangzhou International Bio-Island, Guangzhou 510005, China; liang_taizhen@gzlab.ac.cn (T.L.); xiao_shiqi@gzlab.ac.cn (S.X.); liu_sen@gzlab.ac.cn (S.L.); hu_meilin@gzlab.ac.cn (M.H.); li_guojie@gzlab.ac.cn (G.L.); li_peiwen@gzlab.ac.cn (P.L.); 2State Key Laboratory of Respiratory Disease, Guangzhou Medical University, Guangzhou 511400, China; 3School of Pharmaceutical Sciences, Southern Medical University, Guangzhou 510515, China; wuzy913@163.com; 4School of Medicine, South China University of Technology, Guangzhou 510006, China; 15735928408@163.com; 5School of Biology and Biological Engineering, South China University of Technology, Guangzhou 510006, China; 6Zhongshan School of Medicine, Sun Yat-sen University, Guangzhou 510080, China

**Keywords:** SARS-CoV-2, phenothiazine, entry inhibitor, spike, cathepsin L

## Abstract

Novel coronavirus disease 2019 (COVID-19), a respiratory disease caused by severe acute respiratory syndrome coronavirus 2 (SARS-CoV-2), has brought an unprecedented public health crisis and continues to threaten humanity due to the persistent emergence of new variants. Therefore, developing more effective and broad-spectrum therapeutic and prophylactic drugs against infection by SARS-CoV-2 and its variants, as well as future emerging CoVs, is urgently needed. In this study, we screened several US FDA-approved drugs and identified phenothiazine derivatives with the ability to potently inhibit the infection of pseudotyped SARS-CoV-2 and distinct variants of concern (VOCs), including B.1.617.2 (Delta) and currently circulating Omicron sublineages XBB and BQ.1.1, as well as pseudotyped SARS-CoV and MERS-CoV. Mechanistic studies suggested that phenothiazines predominantly inhibited SARS-CoV-2 pseudovirus (PsV) infection at the early stage and potentially bound to the spike (S) protein of SARS-CoV-2, which may prevent the proteolytic cleavage of the S protein, thereby exhibiting inhibitory activity against SARS-CoV-2 infection. In summary, our findings suggest that phenothiazines can serve as a potential broad-spectrum therapeutic drug for the treatment of SARS-CoV-2 infection as well as the infection of future emerging human coronaviruses (HCoVs).

## 1. Introduction

The severe acute respiratory syndrome coronavirus 2 (SARS-CoV-2) is an enveloped positive-stranded RNA virus and a member of the *Betacoronavirus* genus of the Coronaviridae family, which shares high genetic sequence identity with severe acute respiratory syndrome coronavirus (SARS-CoV) of 2003 [1]. SARS-CoV-2 infection can cause a life-threatening respiratory disease called coronavirus disease 2019 (COVID-19). As of May 2023, SARS-CoV-2 has caused over 765 million infection cases, with more than 6.9 million deaths (https://covid19.who.int/, accessed: 30 May 2023). Despite the preliminary success of anti-SARS-CoV-2 drugs and vaccines, continuous viral mutants, especially the newly emerged Omicron variant and its sublineages, bring substantial challenges to the effectiveness of early strain-derived vaccines. Moreover, SARS-CoV-2 is likely to exist within humans for many years, highlighting the critical importance of developing more effective antiviral drugs or vaccines [2].

To date, there are two main classes of antiviral drugs adopted for treating COVID-19 by targeting viral spike (S) protein: RNA-dependent RNA polymerase (RdRp) and main protease (M^pro^) [3]. The first class contains several small-molecule drugs, such as nirmatralvir/ritonavir (paxlovid), molnupiravir, favipiravir, and remdesivir, which have been approved to treat COVID-19 patients [4,5,6]. Molnupiravir, favipiravir, and remdesivir are nucleoside analog drugs that target the RdRp of SARS-CoV-2 to compete with nucleosides to interact with mRNA, and thus disrupt viral RNA replication. However, there are still controversies about their clinical results and effectiveness against SARS-CoV-2, such as recent trials showing that remdesivir had no significant effect on patients with COVID-19 [7]. Moreover, nucleoside analog drugs, such as molupiravir, which enhance the frequency of viral RNA mutagenesis, may possess host mutational activity as well [8]. Furthermore, recent studies found that the nucleoside analog drugs targeting RdRp were susceptible to drug resistance [7,9]. A possible explanation is that mutations at RdRp cause conformational changes of the RdRp active site, which cripples the binding affinities of drugs to RdRp. Recently, paxlovid, the M^pro^ inhibitor, holds the most promise to treat COVID-19 with higher efficacy and oral bioavailability than other clinical administration drugs [10]. However, a few adverse events, such as diarrhea, dysgeusia, and vomiting, occur in recipients of paxlovid [11]. Of note, further investigation found that the M^pro^ inhibitor also tended to be drug-resistant due to the high mutational frequencies of the active-site and dimer-interface hotspots of SARS-CoV-2 M^pro^ [12,13]. The second class belongs to neutralizing antibodies (nAbs), which generally target the spike receptor-binding domain (RBD). However, the RBD of CoV is a highly mutable region along with viral evolution, which leads to the attenuated neutralization activities of nAbs against the emerging Omicron subvariants. Considering continuously emerging escape mutants, ongoing reinfection, and current drug resistance, it is still an urgent need to assess broad-spectrum antiviral drugs for treating COVID-19.

There are now seven pathogenic human coronaviruses. Four low-pathogenic coronaviruses, which include HCoV-NL63, HCoV-OC43, HCoV-229E, and HCoV-HKU1, induce only mild upper respiratory disease. Three highly pathogenic coronaviruses, which include SARS-CoV, MERS-CoV, and SARS-CoV-2, cause severe respiratory syndrome in humans [14]. SARS-CoV-2 contains four major structural proteins, including nucleocapsid (N), membrane (M), envelop (E), and spike (S) proteins. The life cycle of SARS-CoV-2 is initiated by the attachment of S proteins to the host receptor angiotensin-converting enzyme 2 (ACE2) and the subsequent fusion of viral membrane and cellular membrane. Therefore, the trimeric transmembrane S glycoproteins of SARS-CoV-2, which are located on the surface of the virion, play crucial roles during viral attachment, fusion, and entry. Targeting the S protein can directly block viral entry and appears to be the most efficient way to combat virus infection and spread. Thus, the S protein is considered to be an attractive target for developing therapeutic drugs. Compared to nAbs, small molecular inhibitors targeting S protein with oral bioavailability show great potential to confront SARS-CoV-2. Until now, many molecules targeting the S protein have been found to effectively inhibit viral entry. The peptides EK1 and EK1C4 derived from the heptad repeat 1 (HR1) in S2 subunit of the S protein show potent antiviral activity against multiple human coronaviruses (HCoVs) via interfering with the formation of six-helix bundle (6-HB), indicating the potential of targeting S protein to develop broad-spectrum antiviral drugs [15,16]. However, the peptides generally possess a short half-life and poor oral bioavailability, which limits their clinical application. Our group previously reported that the glycopeptide antibiotic, teicoplanin, shows potent efficiency in inhibiting SARS-CoV-2, SARS-CoV, and MERS-CoV by inhibiting the proteolytic activity of cathepsin L (CTSL) on S proteins [17], indicating that inhibitors preventing the proteolytic cleavage of S protein may be promising candidates for the development of broad-spectrum antiviral therapies.

In this study, we identified that the antipsychotic drugs phenothiazines, including perphenazine, fluphenazine decanoate, acepromazine maleate, prochlorperazine maleate, and alimemazine hemitartrate, exhibit potent inhibition against all current SARS-CoV-2 variants and the divergent betacoronaviruses including SARS-CoV and MERS-CoV. Mechanistically, phenothiazines predominantly inhibit SARS-CoV-2 pseudovirus (PsV) infection at the early stage and potentially bind to the S protein of SARS-CoV-2, which may prevent the proteolytic cleavage of the S protein, thereby exhibiting inhibitory activity against SARS-CoV-2 infection.

## 2. Materials and Methods

### 2.1. Cell Lines and Compounds

The human embryonic kidney cell line 293T (HEK293T) and human hepatoma Huh 7 cell line were purchased from American Type Culture Collection (ATCC). The HEK293T-hACE2 cell line was established by infecting HEK293T cells with lentiviruses that expressed hACE2. All the cell lines were cultured in Dulbecco’s modified Eagle’s medium (DMEM, Gibco, NY, USA) supplemented with 10% fetal bovine serum (FBS), 100 units/mL penicillin, 100 μg/mL streptomycin, and 2% L-glutamine (Gibco). All cells have been regularly monitored for detecting mycoplasma by a PCR-based assay and confirmed to be mycoplasma-free.

The compounds including perphenazine, fluphenazine decanoate, acepromazine maleate, prochlorperazine maleate, and alimemazine hemitartrate were purchased from TargetMol (Shanghai, China) and dissolved in dimethylsulfoxide (DMSO) and stored at −80 °C.

### 2.2. Plasmids and Pseudovirus Production

The spike-expressing plasmids including pcDNA3.1-SARS-CoV-2-S-WT (D614; Wuhan-Hu-1), pcDNA3.1-SARS-CoV-2-S-Alpha (B.1.1.7), pcDNA3.1-SARS-CoV-2-S-Beta (B.1.351), pcDNA3.1-SARS-CoV-2-S-Gamma (P.1), pcDNA3.1-SARS-CoV-2-S-Epsilon (B.1.429), pcDNA3.1-SARS-CoV-2-S-lota (B.1.526), pcDNA3.1-SARS-CoV-2-S-Kappa (B.1.617.1), pcDNA3.1-SARS-CoV-2-S-Delta (B.1.617.2), pcDNA3.1-SARS-CoV-2-S-Lambda (C.37), pcDNA3.1-SARS-CoV-2-S-Omicron (BA.1, BA.2, BA.2.12.1, BA.4, XBB, and BQ.1.1), pcDNA3.1-SARS-CoV-S, and pcDNA3.1-MERS-CoV-S were synthesized or preserved in our laboratory.

To produce pseudovirions, HEK293T cells were co-transfected with a plasmid expressing S protein as described above, a backbone plasmid pHIV-Luciferase (Addgene plasmid #21375) that encoded luciferase reporter under the control of EF-1α promoter, and a packaging construct psPAX2 (Addgene plasmid #12260). The cell culture supernatants containing the released virions were collected at 72 h post-transfection, passed through a 0.45 μΜ filter, and frozen at −80 °C.

### 2.3. Luciferase Assay on Pseudovirus Infection

HEK293T-hACE2 cells were seeded into 96-well plates at 5 × 10^4^ cells/well and incubated overnight at 37 °C. The candidate compounds were serially diluted and mixed with an equal volume of respective pseudoviruses and then incubated at 37 °C for 30 min. The mixture was transferred to the HEK293T-hACE2 cells and incubated for 48 h. Then, the cells were lysed with lysis buffer, and the luciferase activity was assessed using Luciferase Assay Kits. The IC_50_ was calculated as the final concentration of the candidate compound that caused a 50% reduction in relative luminescence units (RLUs) compared to the level of the virus control subtracted from that of the cell control.

### 2.4. Cytotoxicity Assay

HEK293T-hACE2 cells were seeded into 96-well plates at 2 × 10^4^ cells/well and incubated overnight at 37 °C. The candidate compounds were serially diluted and added to the cells. DMSO was used as the negative control. After 48 h post-incubation, about 10 μL of Cell Counting Kit 8 (WST-8/CCK-8) solution, which uses a water-soluble tetrazolium salt to quantify the number of live cells by producing an orange formazan dye upon bio-reduction, was added to each well and incubated at 37 °C for another 4 h. Then, the optical density (OD) indicating live cells was measured at 570 nm by a microplate reader. The 50% cytotoxicity concentrations (CC_50_) were calculated using Prism 8.0 (San Diego, CA, USA).

### 2.5. Time-of-Addition Assay

HEK293T-hACE2 cells were seeded into 96-well plates at 2 × 10^4^ cells/well and incubated overnight at 37 °C. The cells were incubated with SARS-CoV-2 PsV (D614), while the candidate compounds at the indicated concentration were added 0.5 h before or 0, 0.5, 1, 2, 4, 6, or 8 h after the addition of the PsV. Cells were lysed 48 h post-infection to determine the entry inhibition efficacy.

### 2.6. Drug or Virus Pretreatment Assay

To figure out whether the target of phenothiazines was the virus or host cell to block the entry of SARS-CoV-2, we performed three kinds of treatment assays directed toward the host cell or virus. For the drug pretreatment group: to examine whether the phenothiazines could block the viral receptor to inhibit viral attachment to the host cells or if it could induce production of antiviral host factors, HEK293T-hACE2 cells were incubated with candidate compounds including perphenazine, fluphenazine decanoate, acepromazine maleate, prochlorperazine maleate, and alimemazine hemitartrate at the indicated concentration at 37 °C for 1 h, and then washed with DMEM twice before the addition of SARS-CoV-2 PsV. For the drug-virus co-treatment group: to examine whether the phenothiazines could bind to the S protein to block viral attachment or fusion, candidate compounds were co-incubated with SARS-CoV-2 PsV at 37 °C for 1 h and then co-treated the HEK293T-hACE2 cells. For the virus pretreatment group: to determine whether the antiviral effect of phenothiazines was the post-entry steps, such as genome translation and replication, virion assembly, and virion release from the cells, HEK293T-hACE2 cells were pre-infected with SARS-CoV-2 PsV for 1 h and then washed with DMEM twice before the addition of different concentrations of compounds. All of the cells were lysed 48 h post-infection and measured for the amounts of luciferase.

### 2.7. Surface Plasmon Resonance (SPR) Binding Analysis

The interaction affinities between SARS-CoV-2 S protein and candidate compounds were analyzed by the Biacore 8K^+^ instrument (Cytiva, MA, USA) at 25 °C in running buffer (20 mM phosphate buffer, 2.7 mM KCl, 137 mM NaCl, 0.05% Surfactant P20 and 5% (*v*/*v*) DMSO). Briefly, SARS-CoV-2 S protein (flow cell 2) was immobilized onto a CM7 sensor chip using a Biocore 8K instrument. A reference surface was generated in another flow channel (flow cell 1) with unloaded S protein. Subsequently, serial dilution of candidate compounds ranging in concentrations from 7.18 to 250 μM was injected over flow cells 1 and 2 at a flow rate of 30 μL/min with a contact time of 120 s and dissociation time of 60 s at 25 °C. Three separate buffer blank injections were run per interaction to provide blanks for double-referencing the data. Compound injections were referenced to a blank surface and by a buffer blank. The response units (RU) were calculated after subtracting the response of the reference channel (flow cell 1) without spike protein from the responses of active (flow cell 2) prior to data analysis, and the resulting data were fitted to a 1:1 binding model using Biacore insight Evaluation Software (v4.0.8.19879; Cytiva).

### 2.8. CTSL-Mediated the Cleavage of SARS-CoV-2 Spike Protein In Vitro

To determine whether the phenothiazine would affect the cleavage of SARS-CoV-2 S by the CTSL protein (Sino Biological, Beijing, China, 10486-H08H), about 2 μg recombinant SARS-CoV-2 S protein (Sino Biological, 40589-V08B1) was incubated with the candidate compound on ice for 15 min. Additionally, about 10 ng recombinant CTSL protein was incubated with or without teicoplanin in the CTSL activation buffer (400 mM sodium acetate, 4 mM EDTA, 8 mM DTT and pH 5.5), and the mixture was incubated on ice for 15 min for pre-activation. After incubation, the phenothiazine-S mixture was added to the pre-activated CTSL protein solution and incubated at 37 °C for 1 h. Finally, all of the samples were proceeded to the sodium dodecyl sulfate–polyacrylamide gel electrophoresis (SDS-PAGE) and analyzed with silver staining (PROTSIL2-1KT, Sigma-Aldrich, St. Louis, MO, USA) according to the manufacturer’s instructions.

### 2.9. The Cleavage of SARS-CoV-2 Spike Protein

For the SARS-CoV-2 spike cleavage assay, prochlorperazine maleate or teicoplanin was co-incubated with SARS-CoV-2 PsV at 37 °C for 1 h prior to being added to the HEK293T-hACE2 cells. After 12 h of treatment, the cells were collected and washed with PBS two times, followed by lysing by NP-40 lysis buffer (10 mM Tris-HCl, 150 mM NaCl, 0.5% NP-40, 1% TritonX-100, 10% glycerol, 2 mM EDTA, 1 mM NaF, and 1 mM Na3VO4) on ice for 30 min. After centrifugation for 10 min at 12,000× *g* for 10 min at 4 °C, the supernatant was collected and boiled in SDS sample loading buffer and analyzed by SDS-PAGE. Specific primary antibodies were used, followed by the use of the secondary antibody.

### 2.10. Antiviral Experiments In Vivo

K18-hACE2 transgenic mice (6–7 weeks old) were purchased from GemPharmatech Co., Ltd. (Nanjing, China) and housed in SPF conditions. All the animal procedures were carried out in accordance with the guideline of ARRIVE (Animal Research: Reporting of In Vivo Experiments) and approved by the Ethics Committee of Guangzhou National Laboratory.

To test the antiviral effects of phenothiazine *in vivo*, the K18- hACE2 transgenic mice were randomized into two groups (five mice per group): physiological saline (vehicle control) and prochlorperazine maleate (10 mg/kg). The animal grouping and treatment are schematically illustrated in Figure 5A. Briefly, mice were administered 10 mg/kg body weight of prochlorperazine maleate or physiological saline via intraperitoneal (i.p.) injection. One day after the drug treatment, all the mice were intranasally (i.n.) challenged with SARS-CoV-2 spike-pseudovirus (D614) containing the firefly luciferase gene, which was produced as above described. Thereafter, mice received a half dose of the first treatment orally with physiological saline (vehicle control) or 5 mg/kg prochlorperazine maleate up to day 4 post-infection. At day 5 post-infection, mice were sacrificed, and the lung tissues were collected for determination of p55 and p24 levels by Western blotting and relative *luciferase* mRNA levels by RT-qPCR.

### 2.11. Real-Time Quantitative PCR (RT-qPCR)

Total RNA was extracted from the lungs of the mice using Trizol (Invitrogen, Carlsbad, CA, USA) according to the manufacturer’s instructions and reverse transcribed with HiScript II RT SuperMix (Vazyme, Nanjing, China). PCR was performed using ChamQ Universal SYBR qPCR Master Mix (Vazyme, China) on a LightCycler 480 under the following conditions: 95 °C for 30 s for initial denaturation, followed by 40 cycles of 95 °C for 10 s and 60 °C for 30 s for annealing and extension, respectively. The mRNA levels of *Luciferase* were normalized to the *Hypoxanthine guanine phosphoribosyltransferase* (*HPRT*) gene as the reference control. Relative gene expression levels were determined using the 2^−ΔΔCT^ method. The sequences of the primers used for PCR amplification are listed as follow: HPRT Forward: 5′-AGTCCCAGCGTCGTGATTAG-3′; HPRT Reverse: 5′-TGGCCTCCCATCTCCTTCAT-3′; Luciferase Forward: 5′-GTGCAGCGAGAATAGCTTGC-3′; Luciferase Reverse: 5′-TTGCTCACGAATACGACGGT-3′.

### 2.12. Statistical Analysis

Experiment data are presented as mean ± SD, which indicated at least three replicates. Statistical analysis was calculated with GraphPad Prism 8.0 software (San Diego, CA, USA). One-way ANOVA followed by Dennett’s multiple comparison post hoc test was used to detect differences between multiple groups. *p*-values below 0.05 (*p* < 0.05) were considered to demonstrate statistically significant differences. Statistical significance was defined as * *p* < 0.05, ** *p* < 0.01, *** *p* < 0.001.

## 3. Results

### 3.1. Phenothiazines Show Potent Broad-Spectrum Antiviral Activities

To discover broad-spectrum antiviral drugs for SARS-CoV-2, we first tested a panel of a 1600-member FDA-approved drug library (Catalog No. L4200; Topscience, Shanghai, China) to identify potential entry inhibitors using SARS-CoV-2 Omicron BA.4 subvariant pseudovirus (PsV) as we previously described [18]. Briefly, the candidate drugs were mixed with an equal volume of SARS-CoV-2 Omicron BA.4 PsV and then incubated at 37 °C for 30 min. The mixture was transferred to the HEK293T-hACE2 cells and incubated for 48 h. Then, the cells were lysed with lysis buffer, and the luciferase activity was assessed using Luciferase Assay Kits. To exclude the drugs that only inhibited early events of the HIV-1 life cycle and to identify SARS-CoV-2-S-specific drugs, HIV-luc/VSV-G pseudotyped viruses bearing vesicular stomatitis virus (VSV) glycoproteins were used for secondary screening of the initial screening compounds. The screening experiments using the library of FDA-approved drugs against SARS-CoV-2 Omicron (BA.4) PsV are schematically illustrated in Figure 1A. As shown in Figure 1B, we found that perphenazine (chemical structure in Figure 1C) and acepromazine maleate (chemical structure in Figure 1D) possessed significant inhibitory activities, which were comparable to that of hydroxychloroquine (HCQ), the positive control (Figure 1B). Then, we validated the antiviral activities of perphenazine and acepromazine maleate by using SARS-CoV-2 early strain-derived PsV. Results showed that perphenazine (Figure 1C) and acepromazine maleate (Figure 1D) exhibited potent inhibitory activities against the entry of SARS-CoV-2 S (D614) PsV to the HEK293T-hACE2 cells (HEK293T cells stably expressing hACE2) in a dose-dependent manner, with IC_50_ of 0.831 μM and 1.823 μM, respectively.

Since both perphenazine and acepromazine maleate belong to the phenothiazine compounds, we speculated that other phenothiazines might also exhibit antiviral activities against SARS-CoV-2. To this aim, we further explored the antiviral effects of other phenothiazines including fluphenazine decanoate (chemical structure in Figure 1E), prochlorperazine maleate (chemical structure in Figure 1F), and alimemazine hemitartrate (chemical structure in Figure 1G) on SARS-CoV-2 S (D614) PsV. Surprisingly, all the tested phenothiazines showed potent inhibition activities, with IC_50_ of 3.903 μM (Figure 1E), 0.259 μM (Figure 1F), and 0.477 μM (Figure 1G), respectively. To evaluate the toxicities of phenothiazines, HEK293T-hACE2 cells were treated with serial concentrations of candidate compounds and assayed by CCK-8. As shown in Table 1, all of the candidate phenothiazines displayed minimal cellular toxicities, with CC_50_ values ranging from 18.120 μM to 93.300 μM. The CC_50_ values of the candidate phenothiazines were more than 15 times higher than their IC_50_ values for inhibiting SARS-CoV-2 S (D614) PsV. Their selectivity indexes (SI = CC_50_/IC_50_) ranged from 15.775 to 328.803. These results demonstrated that phenothiazines might be safe to be treated as anti-SARS-CoV-2 drugs for clinical use.

To investigate whether the tested phenothiazines had the same effects on SARS-CoV, which is closely related to SARS-CoV-2 and also exploits ACE2 for cell entry, we conducted a pilot experimental test *in vitro* to evaluate the anti-SARS-CoV PsV activity. Consistently, the data showed that all of the tested phenothiazines including perphenazine, acepromazine maleate, fluphenazine decanoate, prochlorperazine maleate, and alimemazine hemitartrate also showed significant efficacies in inhibiting SARS-CoV PsV infection with IC_50_ of 1.159 μM (Figure 1C), 1.391 μM (Figure 1D), 3.955 μM (Figure 1E), 0.773 μM (Figure 1F), and 0.356 μM (Figure 1G), respectively. Notably, all of the tested phenothiazines also showed similar potency to inhibit the entry of MERS-CoV PsV, but with no inhibitory activity against the control vesicular stomatitis virus (VSV) PsV (Figure 1), indicating the antiviral activity of phenothiazines is specific for HCoVs surface glycoproteins. Furthermore, we also found that phenothiazines at the concentration of 10 μM significantly inhibited SARS-CoV-2 S-mediated cell–cell fusion (Appendix A). Taken together, these data indicated that phenothiazines might be safe and potent broad-spectrum antiviral drugs for human pathogenic coronavirus diseases.

### 3.2. Phenothiazines Inhibit the Entry of Multiple SARS-CoV-2 Variants

Since December 2019, SARS-CoV-2 (D614) has continuously expanded into extensive variants or sublineages, including B.1.1.7 (Alpha), B.1.351 (Beta), P.1 (Gamma), B.1.429 (Epsilon), B.1.526 (lota), B.1.617.1 (Kappa), B.1.617.2 (Delta), and C.37 (Lambda), which seriously threaten human health. To further assess the efficacy of phenothiazines on other circulating strains, we constructed several S-expressing plasmids that were derived from various SARS-CoV-2 mutants. All of the tested phenothiazines showed potent inhibitory activities against B.1.1.7 (Alpha), B.1.351 (Beta), P.1 (Gamma), B.1.429 (Epsilon), B.1.526 (lota), B.1.617.1 (Kappa), B.1.617.2 (Delta), and C.37 (Lambda) PsVs, with IC_50_ ranging from 0.315–1.315 μM for perphenazine (Table 2 and Appendix A), 1.200–9.318 μM for fluphenazine decanoate (Table 2 and Appendix A), 0.661–4.605 μM for acepromazine maleate (Table 2 and Appendix A), 0.113–0.762 μM for prochlorperazine maleate (Table 2 and Appendix A), and 0.293–1.776 μM for alimemazine hemitartrate (Table 2 and Appendix A).

The Omicron variant has become the globally dominant SARS-CoV-2 variant and continuously evolved hundreds of subvariants. We next focused on investigating the inhibitory activities of phenothiazines on diverse Omicron subvariants. As shown in Figure 2, the results also verified the potent activities of the phenothiazines against the infection of diverse Omicron subvariants, including BA.1, BA.2, BA.2.12.1, BA.4, XBB, and BQ.1.1. Perphenazine inhibited the BA.1, BA.2, BA.2.12.1, BA.4, XBB, and BQ.1.1 PsVs, with IC_50_ values of 0.534, 0.540, 0.382, 0.664, 1.915, and 0.890 μM, respectively, on HEK293T-hACE2 cells (Figure 2A). Compared to perphenazine, fluphenazine decanoate (Figure 2B) and acepromazine maleate (Figure 2C) showed less efficacy in inhibiting BA.1, BA.2, BA.2.12.1, BA.4, XBB, and BQ.1.1 PsVs infection, with IC_50_ ranging from 3.724 to 9.687 μM and 1.411 to 3.675 μM, respectively. Prochlorperazine maleate (Figure 2D) and alimemazine hemitartrate (Figure 2E) showed comparable efficacy with perphenazine in inhibiting BA.1, BA.2, BA.2.12.1, BA.4, XBB, and BQ.1.1 PsVs infection, with IC_50_ ranging from 0.289 to 0.804 μM and 0.506 to 1.460 μM, respectively. Notably, the inhibitory activities of all the tested phenothiazines against divergent SARS-CoV-2 variants as well as the recently emerging Omicron subvariants were comparable to those against early strain D614 (Table 1), indicating that the antiviral activity of phenothiazines was not affected by the mutations on the S proteins. Taken together, these results demonstrated that phenothiazines exhibited broad-spectrum antiviral activities.

### 3.3. Phenothiazines Predominantly Inhibit SARS-CoV-2 PsV Infection at the Early Stage

To further decipher the underlying mechanism of phenothiazines against SARS-CoV-2 infection, a time-of-addition assay was conducted to determine which step of SARS-CoV-2 infection was affected by phenothiazines. First, phenothiazines at the indicated concentration were added to the targeted cells HEK293T-hACE2 at different time points, as shown in Figure 3A. As shown in Figure 3B–F, no inhibitory activity was observed when phenothiazines were added 0.5 h before SARS-CoV-2 infection, while all of the tested phenothiazines exhibited more than 70% inhibitory activity against SARS-CoV-2 PsVs infection at 0, 0.5, 1, and 2 h after the addition of viruses. The inhibitory activity was gradually decreased when phenothiazines were added 4 h later (Figure 3B), and the inhibitory activity of prochlorperazine maleate was sharply decreased to about 20% at 8 h after the addition of SARS-CoV-2 PsV (Figure 3E). These results indicated that phenothiazines might target the early stage of the viral entry.

To figure out whether the target of phenothiazines was the virus or host cell to block the entry of SARS-CoV-2, we performed three kinds of treatment assays directed toward the host cell or virus, and the drug treatments are schematically illustrated in Figure 3G. As shown in Figure 3H–L, all of the phenothiazines showed potent inhibitory activity against SARS-CoV-2 PsV infection in a dose-dependent manner when the tested compound and SARS-CoV-2 PsV were co-incubated at 37 °C for 1 h and then added to the cells at the same time (green bar). However, when HEK293T-hACE2 target cells were pre-treated with the tested phenothiazines for 1 h before SARS-CoV-2 PsV, or HEK293T-hACE2 cells were pre-infected with SARS-CoV-2 PsV 1 h before the inhibitor was added, all of the tested phenothiazines showed no inhibitory activities (Figure 3H–L, red and blue bar). These results indicated that the phenothiazines-mediated antiviral activity might be directly attributed to the viral component, especially the S protein, but not the host cell.

### 3.4. Phenothiazines Bind to S Protein and Interfere S Protein Cleavage

Our preliminary experiment has revealed that phenothiazines might be viral entry inhibitors by targeting the spike glycoprotein of coronaviruses. To verify this hypothesis, we performed a surface plasmon resonance (SPR) binding assay to determine the binding affinities of phenothiazines with the SARS-CoV-2 S protein. In this experiment, purified SARS-CoV-2 S protein was immobilized on the CM7 sensor chip, and the candidate compounds with different concentrations (from 7.18 to 250 μM) flowed over it. The results showed that the candidate phenothiazines potentially interacted with the S protein, with an equilibrium dissociation constant (*K_D_*) of 979 μM for perphenazine (Figure 4A), 60.2 μM for fluphenazine decanoate (Figure 4B), 53.1 μM for acepromazine maleate (Figure 4C), 48.4 μM for prochlorperazine maleate (Figure 4D), and 435 μM for alimemazine hemitartrate (Figure 4E), indicating that the candidate phenothiazines targeted the SARS-CoV-2 S protein directly to inhibit viral entry.

It is well known that the trimeric transmembrane S glycoproteins on the surface of SARS-CoV-2 are composed of a receptor-binding unit S1 and a membrane-fusion unit S2. During infection, the S protein needs to be cleaved to proteolytic activation by the host cell proteases such as CTSL, which changes the conformation of the S protein from the pre-fusion state to the post-fusion state and eventually favors membrane fusion and genome release [19,20,21,22]. To evaluate whether the interaction of phenothiazine with S protein would affect S protein proteolytic cleavage, we herein performed a CTSL enzymatic cleavage assay to examine the proteolysis of the SARS-CoV-2 S protein with or without phenothiazines. As shown in Figure 4F,G, the purified CTSL protein functionally cleaved the SARS-CoV-2 S protein into smaller fragments, with decreasing levels of full-length S protein. However, pre-treatment with phenothiazines including perphenazine, acepromazine maleate, fluphenazine decanoate, prochlorperazine maleate, and alimemazine hemitartrate prevented the cleavage of S protein by CTSL protein, similar to the inhibition of teicoplanin, which inhibits SARS-CoV-2 entry by specifically inhibiting the proteolytic activity of CTSL [17] (Figure 4F,G), although phenothiazines do not completely recover the CTSL-mediated S protein cleavage levels. To provide further evidence for the role of phenothiazines preventing the cleavage of S protein to inhibit SARS-CoV-2 entry, we co-incubated prochlorperazine maleate with SARS-CoV-2 PsV at 37 °C for 1 h and then co-treated the HEK293T-hACE2 cells for 12 h. As shown in Figure 3G, both prochlorperazine maleate and teicoplanin treatment also prevented the cleavage of S protein as higher full-length S protein levels were observed. Taken together, these results further confirm that phenothiazines interacted with the S protein, which might prevent the proteolytic cleavage of the S protein, thus inhibiting the entry of HCoVs. However, phenothiazines may not just prevent the cleavage of CTSL—they may also target other proteases such as TMPRSS2 and furin. In patients with SARS-CoV-2 infection, the most susceptible organ is the lung, which contains many kinds of protease that may facilitate the cleavage-mediated S activation. Thus, further research is warranted to evaluate more detailed mechanisms of phenothiazines to inhibit the entry of SARS-CoV-2 besides inhibiting CTSL cleavage.

### 3.5. Prochlorperazine Maleate Inhibits SARS-CoV-2 PsV Infection in K18-hACE2 Mice

To further determine the *in vivo* antiviral effects of phenothiazine on the pseudovirus invasion, K18-hACE2 transgenic mice that were generated by knocking in the human cytokeratin 18 (K18) promoter-driven human ACE2 within the mouse Hipp11 (H11) “safe-harbor” locus [23] were intraperitoneally injected with 10 mg/kg body weight of prochlorperazine maleate or an equal volume of physiological saline and then intranasally exposed to SARS-CoV-2 pseudoviruses. The SARS-CoV-2 S/HIV-1 pseudoviruses harbored an integrated luciferase gene. Thus, the expression levels of HIV-1 p55, HIV-1 p24, and luciferase could reflect the infectivity of pseudotyped viruses. Infectivity was monitored 5 days post-infection by determination of p55 and p24 protein levels by Western blotting and relative luciferase mRNA levels by RT-qPCR. The animal grouping strategy and treatment are schematically illustrated in Figure 5A. Consistent with our *in vitro* findings, prochlorperazine maleate treatment also showed a significant reduction in the viral load as evidenced by the decreased HIV-1 Gag p55 and p24 protein levels in the tissues of the lung compared with the physiological saline group (Figure 5B–D). Moreover, we also found that prochlorperazine maleate treatment significantly inhibited the infection of pseudotyped SARS-CoV-2 infection in K18-hACE2 transduced mice, as determined by the expression of the luciferase gene in the lung tissues (Figure 5E). These data demonstrated that prochlorperazine maleate treatment was able to prevent the infection of SARS-CoV-2 PsV in K18-hACE2 mice, and further experiments are required to explore the detailed dose–response relationship of these phenothiazines *in vivo*.

**Figure 5 viruses-15-01666-f005:**
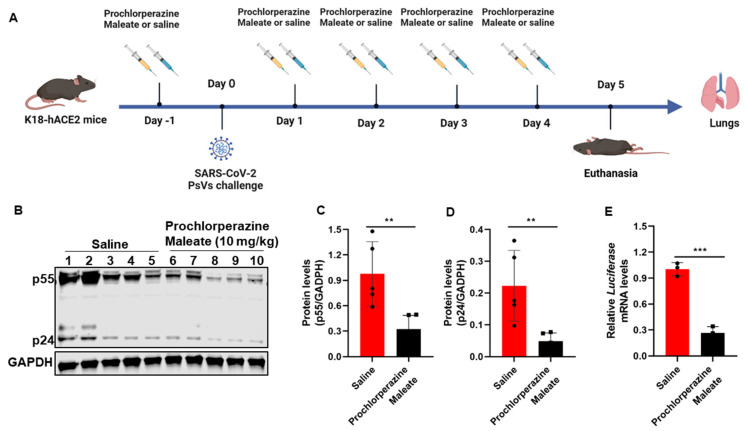
Effect of prochlorperazine maleate against SARS-CoV-2 PsV infection *in vivo*. (**A**) Schematic diagram of the mice challenge experiment. The K18- hACE2 transgenic mice were administered 10 mg/kg body weight of prochlorperazine maleate or an equal volume of physiological saline one day prior to SARS-CoV-2 spike-pseudovirus (D614) infection. Five days post-infection, mice were sacrificed and the lung tissues were collected for determination of p55 and p24 levels by Western blotting and relative *luciferase* mRNA levels by RT-qPCR. (**B**–**D**) The expression levels of Gag p55 and p24 in lung tissues indicated the infection ability of PsV was measured by Western blotting (**B**). The expression of Gag p55 (**C**) and p24 (**D**) protein were quantitated by measuring band intensities using ImageJ software. (**E**) RT-PCR analysis of the mRNA expression of luciferase in lung tissue of mice. Data are presented as means ± SD. *p*-values below 0.05 (*p* < 0.05) were considered to demonstrate statistically significant differences. Statistical significance was defined as ** *p* < 0.01, *** *p* < 0.001.

## 4. Discussion

Currently, the emerging SARS-CoV-2 variants continue to cause waves of new infections globally. Developing effective and broad-spectrum antiviral drugs against SARS-CoV-2 and its variants is an urgent need. Viral entry inhibitors have been proven effective and safe for the treatment of viral infection [24,25]. We previously reported that anhydride-modified proteins serve as a viral entry inhibitor to prevent several kinds of virus infection, including HIV [26,27], influenza [28], HPV [29], RSV [30], MERS-CoV [31], and SARS-CoV-2 [32], indicating that targeting viral entry is considered an attractive drug target for developing broad-spectrum antiviral drugs. We herein report that phenothiazines that have been used in the treatment of schizophrenia and mania in bipolar disorders and psychosis show potent inhibition on the entry of divergent pseudotyped SARS-CoV-2 variants including B.1.1.7 (Alpha), B.1.351 (Beta), P.1 (Gamma), B.1.429 (Epsilon), B.1.526 (lota), B.1.617.1 (Kappa), B.1.617.2 (Delta), C.37 (Lambda), as well as Omicron variants and their sublineages.

To define the molecular mechanism underlying the broad activity of phenothiazines against SARS-CoV-2, we conducted a time-of-addition assay and confirmed that phenothiazine-mediated antiviral activity mainly targeted the early stage of the viral infection. The dopamine receptor D2 antagonist alimemazine, one of the phenothiazines, was previously shown to effectively inhibit the infection of SARS-CoV-2 by interfering with the interaction between the SARS-CoV-2 S protein and neuropilin-1 [33]. However, there are conflicts regarding neuropilin-1 expression in HEK293T cells, indicating that phenothiazines may use other mechanisms to combat SARS-CoV-2 infection. Our further investigation found that phenothiazine-mediated antiviral activity might be directly attributed to the viral component, but not the host cell. Similar to other viruses, the life cycle of SARS-CoV-2 is initiated by the attachment of the viral surface spike glycoprotein to the host cell surface receptor ACE2 through the RBD of spike. Considering the central role of the S protein in viral entry, we verified the interaction between phenothiazines and the S protein. Accordingly, we demonstrated that phenothiazines potentially bound to the S protein and blocked the entry of SARS-CoV-2 PsV into the target cells. The S protein comprises the S1 and S2 subunits and exists in a metastable prefusion conformation. However, upon the receptor binding, the conformational changes of S complex are triggered, which destabilizes the prefusion trimer, resulting in the shedding of the S1 subunit and activating the fusogenic activity of the S2 subunit [20,34]. The interaction of phenothiazines with S protein may prevent the conformational change of S protein, which is essential for the receptor binding and membrane fusion.

Massive studies have reported that phenothiazines exhibit a wide range of antiviral activities, including arenavirus [35], HIV [36], influenza [37], dengue virus (DENV) [38], and coronaviruses [39] by interfering with the binding of the virus to the host cell surface receptor or blocking the endocytosis of the viral particle by inhibiting the required calcium-dependent processes [40]. Interestingly, SARS-CoV-2 also depends on the binding of the virus to the host cell surface receptor or endocytic processes to enter into the target cell. After binding to the ACE2 receptor, the S protein can be proteolytically activated in the presence of the type II transmembrane serine protease (TMPRSS2), and then the activated S protein mediates the fusion of viral membrane and plasma membrane, leading to the release of viral genomes into the cytoplasm and consequential viral replication [41,42]. In the absence of TMPRSS2, the SARS-CoV-2 viral particles enter into the cell through clathrin-mediated endocytosis pathways, which mediate the proteolytic cleavage of S protein by pH-dependent cysteine protease, cathepsin L (CTSL), and eventually resulting in the fusion of viral membrane and the endosome wall [19,21,42]. Interestingly, our results showed that prochlorperazine maleate treatment prevented the cleavage of S protein during spike-mediated pseudovirus entry. Meanwhile, phenothiazines partially recover the levels of full-length S protein, which is processed by CTSL protein-mediated cleavage, indicating that phenothiazines may not just prevent the cleavage of CTSL—they may target other proteases such as TMPRSS2 and furin. In patients with SARS-CoV-2 infection, the most susceptible organ is the lung, which contains many kinds of protease that may facilitate the cleavage-mediated S activation. Thus, further research is warranted to evaluate more detailed mechanisms of phenothiazines to inhibit the entry of SARS-CoV-2 besides inhibiting CTSL cleavage. Theoretically, phenothiazines as a class of compounds seem not that specific as they were also reported to bind numerous other molecular targets such as SARS-CoV-2 M^pro^ [43] and neuropilin-1 [33]. Additionally, a few studies have reported that the potential antiviral effects of phenothiazines may be associated with their ability to inhibit viral binding to plasma membrane receptors [44,45,46]. Therefore, the inconsistency of the binding affinities and the efficacy of inhibiting the S proteolysis of perphenazine and alimemazine hemitartrate may also involve the competitive interaction with other molecular targets. Nevertheless, our study provides a unique insight into phenothiazine derivatives as promising therapeutic drugs for the treatment of SARS-CoV-2 infection by targeting the cleavage of S protein.

Phenothiazines, a class of drugs with antipsychotic properties, have been used in the treatment of schizophrenia and mania in bipolar disorders and psychosis, which benefits from their modulation activities on different neurotransmitter receptors, including histamine H1, dopamine, and serotonergic and cholinergic receptors [47,48]. Therefore, our research raises the possibility that phenothiazines may target additional host receptors or co-receptors, such as dopamine, to modulate cellular entry of SARS-CoV-2. It has been reported that the infection of SARS-CoV-2 is associated with a wide range of neurological manifestations, including encephalopathy, encephalitis, cerebrovascular disease, and headaches [49,50]. Phenothiazines may also be effective in inhibiting SARS-CoV-2 infection in the central nervous system, thus preventing long-term neuronal damage and effectively reducing morbidity. Moreover, most phenothiazines have been approved in clinical application, suggesting that phenothiazines may be safe to treat SARS-CoV-2 infection as well and, hence, may cost less to re-evaluate their toxicities and side effects [47]. Accordingly, we found that all the test compounds displayed low cytotoxicity, with CC_50_ values ranging from 18.120 μM to 93.300 μM, which were over 15 times higher than their IC_50_ for inhibiting SARS-CoV-2 S (D614) PsV.

Due to the incompatibility between SARS-CoV-2 and mouse ACE2 receptor, wild-type mice are less susceptible to SARS-CoV-2. Thus, we conducted BSL2-compliant pseudotyped SARS-CoV-2 infection assays in K18-hACE2 mice that were generated by knocking in the human K18 promoter-driven human ACE2 within the mouse Hipp11 (H11) “safe-harbor” locus [23]. This mice model has been confirmed as a rapid and safe model to better explore viral pathogenesis, assess the effectiveness of drugs *in vivo*, and characterize the potential non-desired effects of these promising drug candidates targeting S-protein [51]. Therefore, we employed a K18-hACE2 mice model to explore the antiviral effect of prochlorperazine maleate candidates that have been validated to be safe for *in vivo* studies [52]. Consistent with our *in vitro* findings, prochlorperazine maleate treatment was also able to prevent the infection of SARS-CoV-2 PsV in K18-hACE2 mice, and further evaluation of the inhibitory effects of prochlorperazine maleate and other phenothiazine candidates against infectious SARS-CoV-2 *in vivo* is warranted.

The global spread of SARS-CoV-2, which causes COVID-19, still threatens human health, and drug development for treating COVID-19 is an urgent need [53]. To date, there are several small molecular drugs that have been approved to treat COVID-19, such as main protease M^pro^ inhibitor nirmatralvir/ritonavir (paxlovid) and RdRp inhibitor molnupiravir, favipiravir, and remdesivir [3,5,10]. Our study verified a novel class of antipsychotic drug phenothiazines that exhibit broad-spectrum activities against SARS-CoV-2 by directly binding S protein to prevent its proteolytic cleavage, suggesting that phenothiazines may have the potential to be used in combination with other approved antiviral drugs with distinct mechanisms of action, such as the M^pro^ inhibitor paxlovid, to produce synergistic effect and raise the genetic barrier to drug resistance. Due to its virulent nature, the SARS-CoV-2 virus is classified as a virus that needs a BSL-3 facility [54]. However, the number of BSL-3 facilities is generally limited due to its demands of negative pressure, tight containment, experienced personnel, and strict laboratory management. Thus, the number of laboratories capable of conducting SARS-CoV-2 virus-based neutralization assay is also limited. Taking all this in consideration, we were unable to determine the efficacy of phenothiazines against authentic SARS-CoV-2 viruses, and the next stage of assessing the efficacy of phenothiazines will be typically involved authentic SARS-CoV-2 viruses and their mutants, humanized animal testing, and synergistic effects with other approved COVID-19 drugs, which is extremely important.

## Figures and Tables

**Figure 1 viruses-15-01666-f001:**
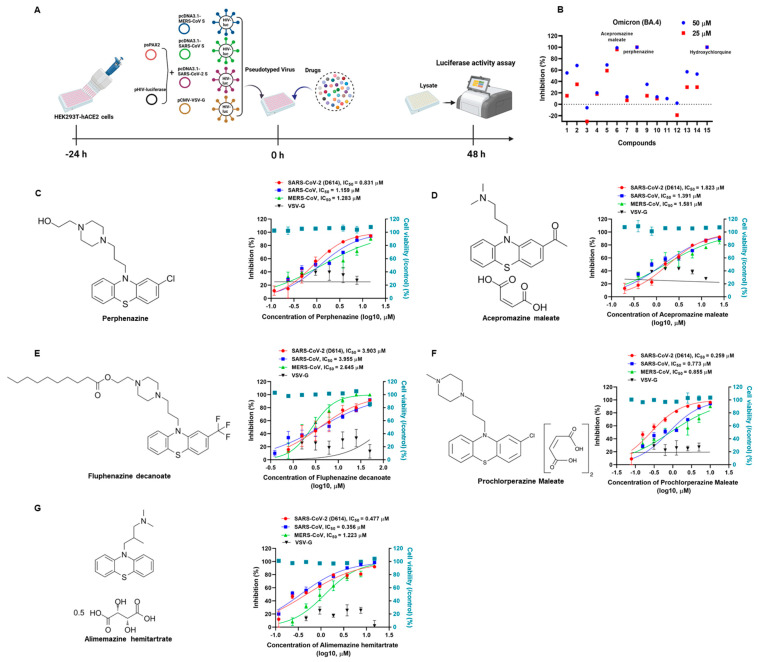
Inhibition of phenothiazines on the infection of SARS-CoV-2, SARS-CoV, and MERS-CoV PsVs. (**A**) Schematic diagram of pseudotype antiviral assay. To produce pseudovirions, psPAX2 plasmids, and pHIV-Luciferase plasmids were co-transfected into HEK293T cells with pcDNA3.1-MERS-CoV S, pcDNA3.1-SARS-CoV S, pcDNA3.1-SARS-CoV-2 S, and pCMV-VSV-G expressing plasmids, respectively. To discover antiviral drugs, HEK293T-hACE2 cells were incubated with candidate compounds and different pseudoviruses. The cells were lysed and the luciferase activity was assessed 48 h post-infection. (**B**) Fourteen drugs were selected after the primary screening experiments using the library of US FDA-approved drugs against SARS-CoV-2 Omicron (BA.4) PsV. The inhibition percentages of selected compounds in the PsV assays are shown as 25 μM (blue dots) or 50 μM (red square dots). Hydroxychloroquine was treated as a positive control. (**C**) The chemical structure of perphenazine and the inhibitory activities of perphenazine against SARS-CoV-2, SARS-CoV, MERS-CoV, and VSV-G PsVs infection. (**D**) The chemical structure of acepromazine maleate and the inhibitory activities of acepromazine maleate against SARS-CoV-2, SARS-CoV, MERS-CoV, and VSV-G PsVs infection. (**E**) The chemical structure of fluphenazine decanoate and the inhibitory activities of fluphenazine decanoate against SARS-CoV-2, SARS-CoV, MERS-CoV, and VSV-G PsVs infection. (**F**) The chemical structure of prochlorperazine maleate and the inhibitory activities of prochlorperazine maleate against SARS-CoV-2, SARS-CoV, MERS-CoV, and VSV-G PsVs infection. (**G**) The chemical structure of alimemazine hemitartrate and the inhibitory activities of alimemazine hemitartrate against SARS-CoV-2, SARS-CoV, MERS-CoV, and VSV-G PsVs infection. In parallel, the toxicities of these compounds in HEK293T-hACE2 cells were analyzed by the CCK-8 assays (blue squares). Samples were tested in triplicate, and the experiments were repeated at least twice.

**Figure 2 viruses-15-01666-f002:**
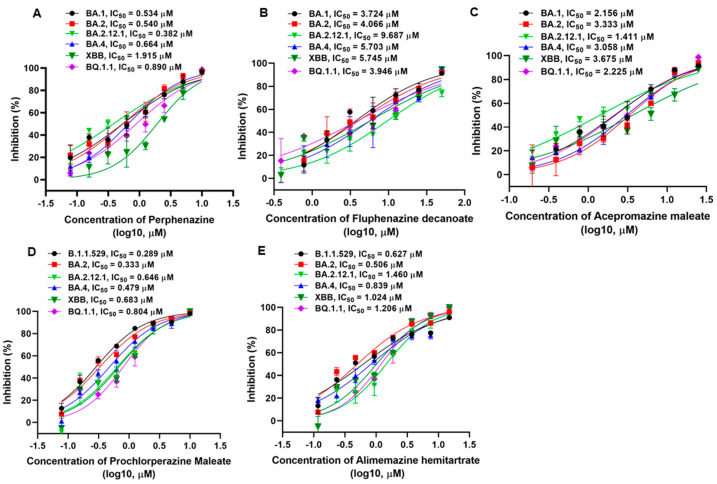
Inhibitory activities of phenothiazines against the infection of pseudotyped SARS-CoV-2 Omicron and its subvariants. (**A**–**E**) Efficacy of perphenazine (**A**), fluphenazine decanoate (**B**), acepromazine maleate (**C**), prochlorperazine maleate (**D**), and alimemazine hemitartrate (**E**) against BA.1, BA.2, BA.2.12.1, BA.4, XBB, and BQ.1.1 PsVs infection. Samples were tested in triplicate, and the experiments were repeated at least twice. Data are presented in mean ± SD.

**Figure 3 viruses-15-01666-f003:**
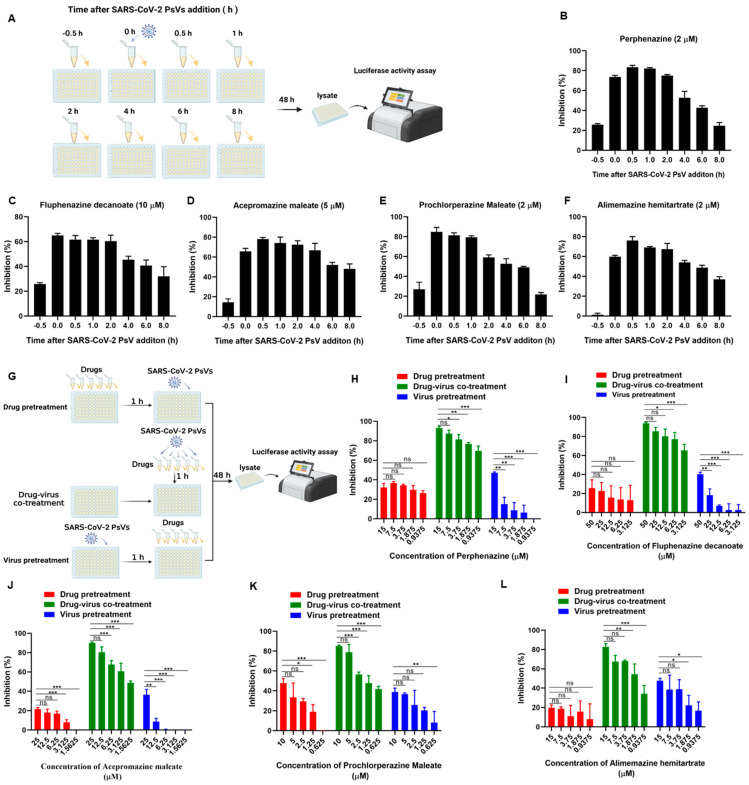
Phenothiazines inhibited SARS-CoV-2 PsV infection at the early stage. (**A**) Schematic diagram of time-of-addition assay. HEK293T-hACE2 cells were incubated with SARS-CoV-2 PsV (D614), and the candidate compounds at the indicated concentration were added 0.5 h before or 0, 0.5, 1, 2, 4, 6, or 8 h after the addition of the PsV. The cells were lysed 48 h post-infection to determine the entry inhibition efficacy. (**B**–**F**) The inhibitory activities of perphenazine (**B**), fluphenazine decanoate (**C**), acepromazine maleate (**D**), prochlorperazine maleate (**E**), and alimemazine hemitartrate (**F**) at the indicated time points before or after SARS-CoV-2 PsV infection. (**G**) Schematic diagram of the three kinds of treatment assays to determine the target of phenothiazines. Drug pretreatment group: HEK293T-hACE2 cells were pre-treated with different concentrations of compounds for 1 h, and then cells were infected with SARS-CoV-2 PsV. Drug-virus co-treatment group: phenothiazines and SARS-CoV-2 PsV were co-incubated at 37 °C for 1 h and then co-treated the cells at the same time. Virus pretreatment group: HEK293T-hACE2 cells were pre-infected with SARS-CoV-2 PsV for 1 h, and then cells were treated with different concentrations of compounds. (**H**–**L**) The inhibitory activities of perphenazine (**H**), fluphenazine decanoate (**I**), acepromazine maleate (**J**), prochlorperazine maleate (**K**), and alimemazine hemitartrate (**L**) against SARS-CoV-2 PsV infection when treated with different ways. Samples were tested in triplicate, and the experiments were repeated at least twice. Data are presented in mean ± SD. One-way ANOVA followed by Dennett’s multiple-comparison post hoc test was used to detect differences between the groups. *p*-values below 0.05 (*p* < 0.05) were considered to demonstrate statistically significant differences. Statistical significance was defined as * *p* < 0.05, ** *p* < 0.01, *** *p* < 0.001.

**Figure 4 viruses-15-01666-f004:**
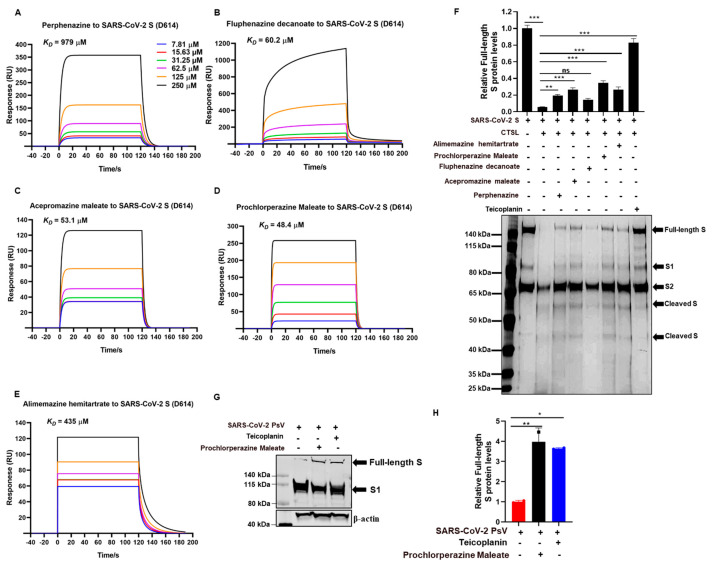
Phenothiazines bound to the spike of SARS-CoV-2 and inhibited its cleavage. (**A**–**E**) Purified SARS-CoV-2 S proteins were immobilized on the CM7 sensor chip, and the candidate compounds including perphenazine (**A**), fluphenazine decanoate (**B**), acepromazine maleate (**C**), prochlorperazine maleate (**D**), and alimemazine hemitartrate (**E**) with different concentrations (from 7.18 to 250 μM) flowed over it for the SPR. The *K_D_* values were analyzed by Biocore Insight Evaluation Software. (**F**) The inhibitory activities of phenothiazines on CTSL-mediated S protein cleavage. Purified CTSL protein was incubated with or without candidate compounds on ice for 15 min, then S protein was added and incubated at 37 °C for 1 h. Proteins were subjected to silver staining. The expression of the full-length protein was quantitated by measuring band intensities using ImageJ software (version 1). Teicoplanin was used as a positive control. (**G**,**H**) Prochlorperazine maleate or teicoplanin was co-incubated with SARS-CoV-2 PsV at 37 °C for 1 h prior to being added to the HEK293T-hACE2 cells. The spike protein level was detected by Western blotting (**G**). The expression of the full-length protein was quantitated by measuring band intensities using ImageJ software (**H**). The values were normalized to β-actin. Data are presented as means ± SD. *p*-values below 0.05 (*p* < 0.05) were considered to demonstrate statistically significant differences. Statistical significance was defined as * *p* < 0.05, ** *p* < 0.01, *** *p* < 0.001.

**Table 1 viruses-15-01666-t001:** The inhibitory activity against SARS-CoV-2 (D614) PsV and cytotoxicity of phenothiazines *in vitro* ^a^.

Compounds	IC_50_ (μM)	CC_50_ (μM)	SI ^b^
**Perphenazine**	0.831 ± 0.120	18.120 ± 1.450	21.805
**Acepromazine maleate**	1.823 ± 0.230	93.300 ± 11.520	51.179
**Fluphenazine decanoate**	3.903 ± 1.090	61.570 ± 28.840	15.775
**Prochlorperazine maleate**	0.259 ± 0.048	85.160 ± 13.915	328.803
**Alimemazine hemitartrate**	0.477 ± 0.124	53.390 ± 10.945	111.929

^a^ Data are presented as mean ± SD of three independent experiments. ^b^ SI, selectivity index = CC_50_/IC_50_.

**Table 2 viruses-15-01666-t002:** Inhibitory activities of phenothiazines against SARS-CoV-2 variant PsVs infection ^a^.

SARS-CoV-2 Variants	IC_50_ (μM, Mean ± SD)				
	Perphenazine	Fluphenazine Decanoate	Acepromazine Maleate	Prochlorperazine Maleate	Alimemazine Hemitartrate
**B.1.1.7 (Alpha)**	1.315 ± 0.283	2.680 ± 0.370	2.580 ± 0.494	0.526 ± 0.103	0.580 ± 0.084
**B.1.351 (Beta)**	1.215 ± 0.225	1.736 ± 0.518	3.152 ± 0.602	0.293 ± 0.027	0.293 ± 0.059
**P.1 (Gamma)**	0.487 ± 0.169	6.214 ± 1.211	1.719 ± 0.348	0.762 ± 0.100	1.526 ± 0.339
**B.1.429 (Epsilon)**	0.802 ± 0.125	1.200 ± 0.425	4.605 ± 1.068	0.113 ± 0.034	0.342 ± 0.103
**B.1.526 (lota)**	0.315 ± 0.042	4.525 ± 2.615	1.298 ± 0.196	0.31 ± 0.045	1.040 ± 0.574
**B.1.617.1 (Kappa)**	0.343 ± 0.075	3.957 ± 0.762	0.661 ± 0.161	0.249 ± 0.057	0.609 ± 0.082
**B.1.617.2 (Delta)**	0.903 ± 0.211	9.318 ± 3.236	2.496 ± 0.583	0.609 ± 0.128	1.776 ± 0.660
**C.37 (Lambda)**	0.482 ± 0.059	2.270 ± 0.419	1.993 ± 0.576	0.423 ± 0.096	0.342 ± 0.097

^a^ The data are presented as mean ± SD of three independent experiments.

## Data Availability

The datasets generated for this study are available on request to the corresponding author.

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
