# Peer review of "Phenothiazines Inhibit SARS-CoV-2 Entry through Targeting Spike Protein"

_viruses, 2023, doi:10.3390/v15081666_

Round 1

Reviewer 1 Report

The manuscript analyzed phenothiazine drug inhibition of SARS-CoV-2 infection using pseudovirus that contained the SARS-CoV-2 S (PsV).  Several reports have showed antiviral effects of some phenothiazines with coronavirus and SARS-CoV-2, and indicated that these drugs can inhibit the main virus protease.  In this manuscript, the authors identified several phenothianizes that inhibited PsV infection of HEK293T-hACE2 cells.  Several tested drugs showed potent PsV inhibition with pseudoviruses having either the S from SARS-CoV-1, MERS-CoV or several SARS-CoV variants, including circulating omicron.  IC50 concentrations were around 1 uM.  The drugs blocked more efficiently PsV infection when administered together with virus particles or soon (up to 2-4 hr) after virus addition to cells, suggesting some effects on S-mediated cell entry.  However, the manuscript did not provide a detailed description of the experiments shown in Figure 3, so that the lower inhibition of the drugs added to cells before the virus could be related to drug degradation, culture washing…(see specific comments below). 

The final Figure 4 shows the analysis of drug binding to the S and drug inhibition of S proteolysis by cathepsin L (CTSL); this is the weakest part of the manuscript.  The binding analysis showed very low drug binding affinity for the S (see specific comments below), significantly lower than the reported IC50s.  In addition, several phenothiazines partially (10-20%) prevented CTSL cleavage of purified S in solution, although with lower activity than the positive control used (80%). Based on these results, the authors concluded that the phenothiazines interacted with the S and prevented its proteolytic cleavage necessary for cell entry. Nonetheless, these drugs can bind also to certain proteases and it is likely that they docked into the CTSL active site and decreased its enzymatic activity. In addition, the data showed very low or negligible drug binding affinity for S, which does not support the conclusion that phenothiazines target the conserved CTSL cleavage sites on the S. Thus, the proposed mechanism for drug blocking of PsV infection is not fully supported by some of the results presented in the last section of the manuscript, and alternative inhibitory processes such as drug targeting of the viral Mpro, cell proteases or cell endocytosis should be considered. Several proteases can prime SARS-CoV-2 virus for cell entry, not just the CTSL.

The use of SARS-CoV-2 S-bearing PsV to identify antiviral drugs is a preliminary approach that needs to be confirmed with SARS-CoV-2 viruses.  Investigation of phenothiazines inhibition of SARS-CoV-2 is necessary to determine their therapeutic potential against COVID-19, which would enhance the scientific interest of this manuscript.

Specific comments:

1.- The authors did not properly describe the methods used to perform the experiments shown in the manuscript.  In particular, the procedures for measuring PsV infection with drug administration at different times shown in Figure 3 must be included in the Methods section.  It is important to know whether the drugs were removed or maintained after virus addition, as well as their concentration throughout the experiment.

2.- The description of the SPR methods was also deficient.  The signals recorded in a control Fc without S were not shown.  It is important to know whether the sensorgrams reported in Figure 4 were corrected for the background signals recorded in a surface without S. The solvent correction must be performed for small molecules with low affinities, such as the drugs analyzed here, for which very high concentrations were used.  In addition, the affinities reported are below the detection limits of some BIAcore instruments (about 50 uM) and the sensorgrams could basically reflect the solvent signal.  The presence of DMSO, used to dissolve the drugs, can also significantly affect the SPR signal.

3.- The KDs determined for the drugs (50-1000 uM) were significantly higher than the IC50 (1 uM) reported in the PsV infection experiments.  This suggested that the SPR experiments were not reliable.

4. The proteolysis experiments in Figure 4F were carried out in vitro with purified S and CTSL.  However, to correlate the effect of the drugs on proteolysis with the infection data shown in the previous figures, I would recommend performing the experiments with cells and PsV and determining the integrity of S by Western blotting.

5.- The manuscript is missing some references to previous studies on phenothiazines and SARS-CoV-2.

Author Response

The manuscript analyzed phenothiazine drug inhibition of SARS-CoV-2 infection using pseudovirus that contained the SARS-CoV-2 S (PsV).  Several reports have showed antiviral effects of some phenothiazines with coronavirus and SARS-CoV-2, and indicated that these drugs can inhibit the main virus protease.  In this manuscript, the authors identified several phenothiazines that inhibited PsV infection of HEK293T-hACE2 cells.  Several tested drugs showed potent PsV inhibition with pseudoviruses having either the S from SARS-CoV-1, MERS-CoV or several SARS-CoV variants, including circulating omicron.  IC50 concentrations were around 1 uM.  The drugs blocked more efficiently PsV infection when administered together with virus particles or soon (up to 2-4 hr) after virus addition to cells, suggesting some effects on S-mediated cell entry.  However, the manuscript did not provide a detailed description of the experiments shown in Figure 3, so that the lower inhibition of the drugs added to cells before the virus could be related to drug degradation, culture washing…(see specific comments below). 

Response: We sincerely apologize for not showing the detailed information of the experiment shown in Figure 3 in our original submission. We have added more detailed information in the Materials and Methods section. Additionally, we added two schematics diagrams in Figure 3 in our revised manuscript to allow readers to more easily comprehend the time points of the methods.

The time-of-addition approach determines how long the addition of a compound can be postponed before it loses its antiviral activity (Dirk Daelemans et al., Nature Protocols, 2011, PMID: 21637207) and has become a well-characterized method to be used to narrow down the mechanism / target of action of a newly discovered anti-viral drug. Therefore, this approach is precise for studying the mechanisms of action of targeted drugs. As described in our revised manuscript, to explore which steps of the viral cycles were blocked by the phenothiazines, a time-of-virus addition experiment was performed. In brief, HEK293T-hACE2 cells were seeded into 96-well plates at 2 х 104 cells/well and incubated overnight at 37℃. The cells were incubated with SARS-CoV-2 PsV (D614), while the candidate compounds including Perphenazine, Fluphenazine decanoate, Acepromazine maleate, Prochlorperazine Maleate and Alimemazine hemitartrate at the indicated concentration were added to the cells at 0.5 hour (h) before or 0, 0.5, 1, 2, 4, 6, or 8 h after addition of the PsV. Cells were lysed 48 h post-infection to determine the entry inhibition efficacy. The time-of-addition assay was schematically illustrated in the newly added Fig. 3A in our revised manuscript. Additionally, the pharmacokinetics of phenothiazines have been widely investigated and the average half-life of Fluphenazine decanoate was 16.4 h, Perphenazine was 9.5 h, Prochlorperazine Maleate was 0.2-13 h, and Alimemazine hemitartrate was 3.6-7 h, indicating that phenothiazines used in our study were steady-state to carry out the time-of-addition assay.

To figure out whether the target of phenothiazines was the virus or host cell to block the entry of SARS-CoV-2, we performed three kinds of treatment assays directed toward the host cell or virus, and the drug treatment procedure was schematically illustrated in newly added Fig. 3G in our revised manuscript. Briefly, for the drug pretreatment group: to examine whether the phenothiazines could block the viral receptor to inhibit viral attachment to the host cells or if it could induce the production of antiviral host factors, HEK293T-hACE2 cells were incubated with candidate compounds including Perphenazine, Fluphenazine decanoate, Acepromazine maleate, Prochlorperazine Maleate and Alimemazine hemitartrate at the indicated concentration at 37°C for 1 h, and then washed with DMEM twice before the addition of SARS-CoV-2 PsV. For the drug-virus co-treatment group: to examine whether the phenothiazines could bind to the S protein to block viral attachment or fusion, candidate compounds were co-incubated with SARS-CoV-2 PsV at 37℃ for 1 h and then co-treated the HEK293T-hACE2 cells. For the virus pretreatment group: to determine whether the antiviral effect of phenothiazines emerged during the post-entry steps, such as genome translation and replication, virion assembly and virion release from the cells, HEK293T-hACE2 cells were pre-infected with SARS-CoV-2 PsV for 1 h and then washed with DMEM twice before the addition of different concentrations of compounds. We have added detailed information in the Materials and Methods section in our revised manuscript.

The final Figure 4 shows the analysis of drug binding to the S and drug inhibition of S proteolysis by cathepsin L (CTSL); this is the weakest part of the manuscript.  The binding analysis showed very low drug binding affinity for the S (see specific comments below), significantly lower than the reported IC50s.  In addition, several phenothiazines partially (10-20%) prevented CTSL cleavage of purified S in solution, although with lower activity than the positive control used (80%). Based on these results, the authors concluded that the phenothiazines interacted with the S and prevented its proteolytic cleavage necessary for cell entry. Nonetheless, these drugs can bind also to certain proteases and it is likely that they docked into the CTSL active site and decreased its enzymatic activity. In addition, the data showed very low or negligible drug binding affinity for S, which does not support the conclusion that phenothiazines target the conserved CTSL cleavage sites on the S. Thus, the proposed mechanism for drug blocking of PsV infection is not fully supported by some of the results presented in the last section of the manuscript, and alternative inhibitory processes such as drug targeting of the viral Mpro, cell proteases or cell endocytosis should be considered. Several proteases can prime SARS-CoV-2 virus for cell entry, not just the CTSL.

Response: Thanks for the reviewer’s instructive questions. If we understand clearly, the reviewer raised two important questions regarding the binding affinities of drugs to S proteins and the inhibition efficiencies of drugs to the cleavage of S. We would like to response to these questions one-by-one.

Firstly, we would like to explain the differences between KD values of SPR results and IC50 values of entry inhibition assays in detail. KD refers to the dissociation rate at equilibrium and is calculated by dividing Koff value by the Kon value. This is the rate at which the small molecule is bound to the target protein (Kon) divided by the rate at which the small molecule unbinds or comes off the target protein (Koff). However, IC50 indeed refers specifically to the inhibition of binding or inhibition of a function at which 50% of the molecule is bound or 50% function is active. Pragmatically, the dissociation constant KD values are generally considered more “constant” and vary less with changes in the concentration of the target protein and molecule. KD values are more independent from the assay setup and type. In contrast, IC50 value can range depending on the assay setup and type (not considered constant values). In some cases, KD and IC50 may be very similar values even identical, but in most experimental conditions, IC50 values do not reflect KD values (Jacques Barbet et al., Pharmaceutical Statistics, 2019, PMID: 30977282). For example, the IC50 values in Fluorescence Polarization depend heavily on what "probe" molecule is used and the concentrations of the target protein and peptide. In our manuscript, the IC50 depends on the titer of the pseudovirus and the concentration of the compounds. In conclusion, the IC50 is not a constant like KD or Ki and cannot be interpreted the same as KD or Ki.

Secondly, we provided more evidence of drug-mediated cleavage inhibition. To elucidate the role of phenothiazine-S protein interaction in the regulation of S protein proteolytic cleavage to inhibit virus entry, we performed additional experiments. To this aim, we co-incubated Prochlorperazine Maleate or teicoplanin with SARS-CoV-2 PsV at 37℃ for 1 h and then co-treated the HEK293T-hACE2 cells for 12 h. As shown in newly added Fig.4 G-H, both Prochlorperazine Maleate and teicoplanin treatment prevented the cleavage of S protein as higher full-length S protein levels were observed. These results further confirmed that phenothiazines interacted with the S protein, which might prevent the proteolytic cleavage of the S protein, thus inhibiting the entry of HCoVs. We also hold a positive viewpoint regarding to the reviewer's opinion that phenothiazines may not just prevent the cleavage of CTSL, they may also target other proteases such as TMPRSS2 and furin. Since in patients with SARS-CoV-2 infection, the most susceptible organ is the lung, which contains many kinds of protease that may facilitate the cleavage-mediated S activation. Thus, further research is warranted to evaluate more detailed mechanisms of phenothiazines to inhibit the entry of SARS-CoV-2 besides inhibiting CTSL cleavage. We have revised the conclusion in the revised manuscript accordingly.

The use of SARS-CoV-2 S-bearing PsV to identify antiviral drugs is a preliminary approach that needs to be confirmed with SARS-CoV-2 viruses.  Investigation of phenothiazines inhibition of SARS-CoV-2 is necessary to determine their therapeutic potential against COVID-19, which would enhance the scientific interest of this manuscript.

Response: We thank the reviewer for the valuable comments. We hold a positive viewpoint regarding assessing the efficacy of phenothiazines against authentic SARS-CoV-2 viruses and their mutants. However, to perform this kind of assay, we have to conduct virus isolation, propagation, and neutralization assay, which must be carried out in the designated facility associated with virus pathogenicity. Due to its virulent nature, the SARS-CoV-2 virus is classified as a virus that needs a BSL-3 facility (Alexa M Kaufer et al., Pathology, 2020, PMID: 33070960). However, the number of BSL-3 facilities is generally limited due to its demands of negative pressure, tight containment, experienced personnel, and strict laboratory management. Thus, the number of laboratories capable of conducting SARS-CoV-2 virus-based neutralization assay is also limited. Taken all consideration, we were unable to determine the efficacy of phenothiazines against authentic SARS-CoV-2 viruses. This issue will be our focus in the next stage of our research and has been described as the limitation of our study within the Discussion section.

However, we assessed the effectiveness of Prochlorperazine Maleate in vivo by employing a K18-hACE2 mice model which was generated by knocking in the human K18 promoter-driven human ACE2 within the mouse Hipp11 (H11) “safe-harbor” locus (Paul B McCray Jr et al., Journal of Virology, 2007, PMID: 17079315). This mice model has been confirmed as a rapid and safe model to better explore the viral pathogenesis, assess the effectiveness of drugs in vivo and characterize their potential non-desired effects of these promising drug candidates targeting S protein (Jiang Chen et al., International Journal of Biological Sciences, 2021, PMID: 34345219). Briefly, K18-hACE2 transgenic mice were intraperitoneally injected with 10 mg/kg body weight of Prochlorperazine Maleate or an equal volume of physiological saline and then intranasally exposed to SARS-CoV-2 pseudoviruses. The SARS-CoV-2 S / HIV-1 pseudoviruses harbored an integrated luciferase gene. Thus, the expression levels of HIV-1 p55, HIV-1 p24 and luciferase could reflect the infectivity of pseudotyped viruses. Infectivity was monitored five days post-infection by determination of p55 and p24 protein levels by Western blotting and relative luciferase mRNA levels by RT-PCR. The animals grouping strategy and treatment were schematically illustrated in newly added Fig. 5A. Consistent with our in vitro findings, Prochlorperazine Maleate treatment also showed a significant reduction of the viral load as evidenced by the decreased HIV-1 Gag p55 and p24 protein levels in the tissues of the lung compared with the physiological saline group (Fig. 5B-D). Moreover, we also found that Prochlorperazine Maleate treatment significantly inhibited the infection of pseudotyped SARS-CoV-2 infection in K18-hACE2 transduced mice, as determined by the expression of the luciferase gene in the lung (Fig. 5E). These data demonstrated that Prochlorperazine Maleate treatment was able to prevent the infection of SARS-CoV-2 PsV in hACE2 mice and further experiments are required to explore the detailed dose-response relationship of these phenothiazines in vivo. We added detailed information to the Results and Discussion in our revised manuscript.

Specific comments:

1.- The authors did not properly describe the methods used to perform the experiments shown in the manuscript.  In particular, the procedures for measuring PsV infection with drug administration at different times shown in Figure 3 must be included in the Methods section.  It is important to know whether the drugs were removed or maintained after virus addition, as well as their concentration throughout the experiment.

Response: We sincerely apologize for not showing the detailed information of the experiment shown in Figure 3 in our original submission. We have added more detailed information in the Materials and Methods section. Additionally, we added two schematics diagrams in Figure 3 in our revised manuscript to allow readers to more easily comprehend the time points of the methods.

As we have replied above, the time-of-addition approach determines how long the addition of a compound can be postponed before it loses its antiviral activity (Dirk Daelemans et al., Nature Protocols, 2011, PMID: 21637207) and has become a well-characterized method to be used to narrow down the mechanism / target of action of a newly discovered anti-viral drug. Therefore, this approach is precise for studying mechanisms of action of targeted drugs. As described in our revised manuscript, to explore which steps of the viral cycles were blocked by the phenothiazines, a time-of-virus addition experiment was performed. In brief, HEK293T-hACE2 cells were seeded into 96-well plates at 2 х 104 cells/well and incubated overnight at 37℃. The cells were incubated with SARS-CoV-2 PsV (D614), while the candidate compounds including Perphenazine, Fluphenazine decanoate, Acepromazine maleate, Prochlorperazine Maleate and Alimemazine hemitartrate at the indicated concentration were added to the cells at 0.5 hour (h) before or 0, 0.5, 1, 2, 4, 6, or 8 h after addition of the PsV. Cells were lysed 48 h post infection to determine the entry inhibition efficacy. The time-of-addition assay was schematically illustrated in newly added Fig. 3A in our revised manuscript. Additionally, the pharmacokinetics of phenothiazines have been widely investigated and the average half-life of Fluphenazine decanoate was 16.4 h, Perphenazine was 9.5 h, Prochlorperazine Maleate was 0.2-13 h, and Alimemazine hemitartrate was 3.6-7 h, indicating that phenothiazines used in our study were steady-state to carry out the time-of-addition assay.

To figure out whether the target of phenothiazines was the virus or host cell to block the entry of SARS-CoV-2, we performed three kinds of treatment assays directed toward the host cell or virus, and the drug treatment procedure was schematically illustrated in newly added Fig. 3G in our revised manuscript. Briefly, for the drug pretreatment group: to examine whether the phenothiazines could block the viral receptor to inhibit viral attachment to the host cells or if it could induce the production of antiviral host factors, HEK293T-hACE2 cells were incubated with candidate compounds including Perphenazine, Fluphenazine decanoate, Acepromazine maleate, Prochlorperazine Maleate and Alimemazine hemitartrate at the indicated concentration at 37°C for 1 h, and then washed with DMEM twice before the addition of SARS-CoV-2 PsV. For the drug-virus co-treatment group: to examine whether the phenothiazines could bind to the S protein to block viral attachment or fusion, candidate compounds were co-incubated with SARS-CoV-2 PsV at 37℃ for 1 h and then co-treated the HEK293T-hACE2 cells. For the virus pretreatment group: to determine whether the antiviral effect of phenothiazines emerged during the post-entry steps, such as genome translation and replication, virion assembly and virion release from the cells, HEK293T-hACE2 cells were pre-infected with SARS-CoV-2 PsV for 1 h and then washed with DMEM twice before the addition of different concentrations of compounds. We have added detailed information in the Materials and Methods section in our revised manuscript.

2.The description of the SPR methods was also deficient. The signals recorded in a control Fc without S were not shown.  It is important to know whether the sensorgrams reported in Figure 4 were corrected for the background signals recorded in a surface without S. The solvent correction must be performed for small molecules with low affinities, such as the drugs analyzed here, for which very high concentrations were used.  In addition, the affinities reported are below the detection limits of some BIAcore instruments (about 50 uM) and the sensorgrams could basically reflect the solvent signal.  The presence of DMSO, used to dissolve the drugs, can also significantly affect the SPR signal.

Response: We sincerely apologize for not showing the important information about the SPR experiment. All of the compound injections indeed have been referenced to a blank surface and by a buffer blank. Actually, the SPR data shown in Figure 4 were already calculated after subtracting the response of the reference channel (flow cell 1) without spike protein from active (flow cell 2) sensorgrams prior to data analysis. We have modified the SPR method in our revised manuscript as below: “The interaction affinities between SARS-CoV-2 S protein and candidate compounds were analyzed by the Biacore 8K+ instrument (Cytiva) at 25℃ in running buffer (20 mM phosphate buffer, 2.7 mM KCl, 137 mM NaCl, 0.05% Surfactant P20 and 5% (v/v) DMSO). Briefly, SARS-CoV-2 S protein (flow cell 2) was immobilized onto a CM7 sensor chip using a Biocore 8K instrument. A reference surface was generated in another flow channel (flow cell 1) with unloaded S protein. Subsequently, serial dilution of candidate compounds ranging in concentrations from 7.18 to 250 μM was injected over flow cells 1 and 2 at a flow rate of 30 μL/min with a contact time of 120 s and dissociation time of 60 s at 25℃. Three separate buffer blank injections were run per interaction, to provide blanks for double-referencing the data. Compounds injections were referenced to a blank surface and by a buffer blank. The response units (RU) were calculated after subtracting the response of reference channel (flow cell 1) without spike protein from the responses of active (flow cell 2) prior to data analysis, and the resulting data were fitted to a 1:1 binding model using Biacore insight Evaluation Software (v4.0.8.19879; Cytiva)”.

The compounds were completely dissolved in the running buffer which contain 5% (v/v) DMSO, and all interaction experiments were performed at 25°C in running buffer (20 mM phosphate buffer, 2.7 mM KCl, 137 mM NaCl, 0.05% Surfactant P20 and 5% (v/v) DMSO). To reduce the effect of DMSO on the SPR signal, we already have set three separate buffer blank injections running per interaction to provide blanks for double-referencing the data, which is also the general process when conducting SPR. Compounds injections were also referenced to a blank surface. In order to detect the binding ability of compounds with S protein more accurate, we try our best to learn the biacore 8K+ instrument and consult the instrument technical consultant, the biacore 8K+ instrument (Cytiva) applied in our experiment do not have the affinity limitation detection as long as the drug is completely dissolved in the solution. Therefore, the results shown in Figure 4 is reliable and we have added more detailed information in the Materials and Methods section in our revised manuscript.

3.- The KDs determined for the drugs (50-1000 uM) were significantly higher than the IC50 (1 uM) reported in the PsV infection experiments.  This suggested that the SPR experiments were not reliable.

Response: We thank the reviewer for these comments. As we have replied above, the differences between KD values of SPR results and IC50 values of entry inhibition assays were explained in detail as follow. KD refers to the dissociation rate at equilibrium and is calculated by dividing Koff value by the Kon value. This is the rate at which the small molecule is bound to the target protein (Kon) divided by the rate at which the small molecule unbinds or comes off the target protein (Koff). However, IC50 indeed refers specifically to the inhibition of binding or inhibition of a function at which 50% of the molecule is bound or 50% function is active. Pragmatically, the dissociation constant KD values are generally considered more “constant” and vary less with changes in the concentration of the target protein and molecule. KD values are more independent from the assay setup and type. In contrast, IC50 value can range depending on the assay setup and type (not considered constant values). In some cases, KD and IC50 may be very similar values even identical, but in most experimental conditions, IC50 values do not reflect KD values (Jacques Barbet et al., Pharmaceutical Statistics, 2019, PMID: 30977282). For example, the IC50 values in Fluorescence Polarization depend heavily on what "probe" molecule is used and the concentrations of the target protein and peptide. In our manuscript, the IC50 depended on the titer of the pseudovirus and the concentration of the compounds. In conclusion, the IC50 is not a constant like KD or Ki and cannot be interpreted the same as KD or Ki.

  1. The proteolysis experiments in Figure 4F were carried out in vitro with purified S and CTSL.  However, to correlate the effect of the drugs on proteolysis with the infection data shown in the previous figures, I would recommend performing the experiments with cells and PsV and determining the integrity of S by Western blotting.

Response: We thank the reviewer for these constructive suggestions. In accordance with the review’s suggestion, we performed additional experiments. Briefly, Prochlorperazine Maleate or teicoplanin was co-incubated with SARS-CoV-2 PsV at 37℃ for 1 h prior added to the HEK293T-hACE2 cells. After 12 h of treatment, the cells were collected and subject to Western blotting. As shown in Fig. 4G-H, Prochlorperazine Maleate and teicoplanin treatment also prevented the cleavage of S protein as higher full-length S protein level observed. These results further confirmed that phenothiazines interacted with the S protein, which may prevent the proteolytic cleavage of the S protein, thus inhibiting the entry of HCoVs. We have added more detailed information in our revised manuscript.

5.- The manuscript is missing some references to previous studies on phenothiazines and SARS-CoV-2

Response: We apologize for missing several references about phenothiazines and SARS-CoV-2. According to the reviewer’s suggestion, we have added some relevant references in our revised manuscript (Rodrigo Machado-Vieira et al., Brazilian Journal of Psychiatry, 2021, PMID: 33440401; M. Plaze et al.,  Encephale, 2020, PMID: 32425222; Jiayu Lu et al., Life sciences, 2021, PMID: 33310043).

Reviewer 2 Report

The data presented in this article contribute to SARS-CoV-2 research through basic research focused on the drug repositioning of phenothiazine derivatives. The authors have described a broad antiviral activity of several compounds mediated by phenothiazines against SARS-CoV-2 and have demonstrated their common molecular mechanism using the SARS-CoV-2 pseudovirus (PsV) infection model, Surface plasmon resonance (SPR), and cleavage of the Spike protein analysis.

The presented data suggest that phenothiazines exhibit potent inhibition against all current SARS-CoV-2 variants, as well as divergent betacoronaviruses, including SARS-CoV and MERS-98 CoV.

However, to manage expectations and highlight the limitations of the study more effectively, it is worth including in the take-home message, even if beyond the scope of this manuscript, that further studies will be necessary to truly consider phenothiazides as a drug against SARS-CoV-2 infection. In vitro infectious studies with the native virus and in vivo studies using infectious models that replicate the clinical and molecular features of COVID-19 disease are essential to validate the findings based on pseudovirus infections in vitro studies. The use of BSL2-compliant pseudo-typed SARS-CoV-2 in K18-hACE2 mice would be indicated to better explore the viral pathogenesis, assess their effectiveness in vivo and caracterize their potential non-desired effects of these promising drug candidates against the S-protein. 

Graphical data representation can be improved, adjustments should be considered before its accept:

Figures needs to be standardized and aligned throughout the manuscript.

Figure 3.

Figure3F-J have very small letters for sttatistics values and box legends letters are not standardized.

Figure 3I is the only one that shows X-axis title in bold.

Appendix B:

Figure S1. Indicate in the panel, eGFP, DAPI and merge pictures.

Figure S2.  All graphs should be aligned and Figure S2-E has a completely different layout than others

Author Response

The data presented in this article contribute to SARS-CoV-2 research through basic research focused on the drug repositioning of phenothiazine derivatives. The authors have described a broad antiviral activity of several compounds mediated by phenothiazines against SARS-CoV-2 and have demonstrated their common molecular mechanism using the SARS-CoV-2 pseudovirus (PsV) infection model, Surface plasmon resonance (SPR), and cleavage of the Spike protein analysis.

The presented data suggest that phenothiazines exhibit potent inhibition against all current SARS-CoV-2 variants, as well as divergent betacoronaviruses, including SARS-CoV and MERS-98 CoV.

However, to manage expectations and highlight the limitations of the study more effectively, it is worth including in the take-home message, even if beyond the scope of this manuscript, that further studies will be necessary to truly consider phenothiazines as a drug against SARS-CoV-2 infection. In vitro infectious studies with the native virus and in vivo studies using infectious models that replicate the clinical and molecular features of COVID-19 disease are essential to validate the findings based on pseudovirus infections in vitro studies. The use of BSL2-compliant pseudo-typed SARS-CoV-2 in K18-hACE2 mice would be indicated to better explore the viral pathogenesis, assess their effectiveness in vivo and caracterize their potential non-desired effects of these promising drug candidates against the S-protein. 

Response: We thanks the reviewer for these valuable comments. As we have replied to another reviewer above, we hold a positive viewpoint regarding to assess the efficacy of phenothiazines against authentic SARS-CoV-2 viruses and their mutants. However, to perform this kind of assay, we have to conduct virus isolation, propagation, and neutralization assay, which must be carried out in the designated facility associated with virus pathogenicity. Due to its virulent nature, the SARS-CoV-2 virus is classified as a virus that needs a BSL-3 facility (Alexa M Kaufer et al., Pathology, 2020, PMID: 33070960). However, the number of BSL-3 facilities is generally limited due to its demands of negative pressure, tight containment, experienced personnel, and strict laboratory management. Thus, the number of laboratories capable of conducting SARS-CoV-2 virus-based neutralization assay is also limited. Taken all consideration, we were unable to determine the efficacy of phenothiazines against authentic SARS-CoV-2 viruses. This issue will be our focus in the next stage of our research, and has been described as the limitation of our study within the Discussion section.

In accordance with the suggestions of the reviewer, we assessed the effectiveness of Prochlorperazine Maleate in vivo by employing a K18-hACE2 mice model which was generated by knocking in the human K18 promoter-driven human ACE2 within the mouse Hipp11 (H11) “safe-harbor” locus (Paul B McCray Jr et al., Journal of Virology, 2007, PMID: 17079315). This mice model has been confirmed as a rapid and safe model to better explore the viral pathogenesis, assess the effectiveness of drugs in vivo and characterize their potential non-desired effects of these promising drug candidates targeting S protein (Jiang Chen et al., International Journal of Biological Sciences, 2021, PMID: 34345219). Briefly, K18-hACE2 transgenic mice were intraperitoneally injected with 10 mg/kg body weight of Prochlorperazine Maleate or an equal volume of physiological saline and then intranasally exposed to SARS-CoV-2 pseudoviruses. The SARS-CoV-2 S / HIV-1 pseudoviruses harbored an integrated luciferase gene. Thus, the expression levels of HIV-1 p55, HIV-1 p24 and luciferase could reflect the infectivity of pseudotyped viruses. Infectivity was monitored five days post-infection by determination of p55 and p24 protein levels by Western blotting and relative luciferase mRNA levels by RT-PCR. The animals grouping strategy and treatment were schematically illustrated in newly added Fig. 5A. Consistent with our in vitro findings, Prochlorperazine Maleate treatment also showed a significant reduction of the viral load as evidenced by the decreased HIV-1 Gag p55 and p24 protein levels in the tissues of the lung compared with the physiological saline group (Fig. 5B-D). Moreover, we also found that Prochlorperazine Maleate treatment significantly inhibited the infection of pseudotyped SARS-CoV-2 infection in K18-hACE2 transduced mice, as determined by the expression of the luciferase gene in the lung (Fig. 5E). These data demonstrated that Prochlorperazine Maleate treatment was able to prevent the infection of SARS-CoV-2 PsV in hACE2 mice and further experiments are required to explore the detailed dose-response relationship of these phenothiazines in vivo. We added detailed information to the Results and Discussion in our revised manuscript.

Graphical data representation can be improved, adjustments should be considered before its accept:

Response: We thank the reviewer for the insightful comments which enabled us to improve our manuscript. According the reviewer’s suggestion, we have checked and redrawn the graphs accordingly to ensure that the manuscript is suitable to be published.

Figures need to be standardized and aligned throughout the manuscript. Figure3F-J have very small letters for statistics values and box legends letters are not standardized.

Response: We sincerely apologize for the graphs which were not aligned. In our revised manuscript, we have redrawn the graphs accordingly to ensure that the manuscript is suitable to be published.

Figure 3I is the only one that shows X-axis title in bold.

Response: We thank the reviewer for pointing out this omission and we have redrawn the figure in the revised manuscript.

Appendix B:

Figure S1. Indicate in the panel, eGFP, DAPI and merge pictures.

Response: We appreciate for the reviewer’s suggestion. Accordingly, we added the indicate in the panel of Figure S1.

Figure S2.  All graphs should be aligned and Figure S2-E has a completely different layout than others

Response: We sincerely apologize for the graphs which were not aligned. In our revised manuscript, we have redrawn the graphs accordingly to ensure that the manuscript is suitable to be published.

Reviewer 3 Report

In their manuscript, the Authors report on antiviral activities of Phenothiazines against SARS-CoV-2. This is a sound piece of work described in appropriate manner. I recommend publication of this material, but have a few comments:

1) What is CCK-8 ? It should be mentined in the text

2) Line 186: „To discover broad-spectrum antiviral drugs for SARS-CoV-2, we firstly tested a panel of US FDA-approved drugs to identify potential entry inhibitors using SARS-CoV-2 Omi cron BA.4 subvariant pseudovirus (PsV).” The exact way in which phenothothiazines were first identified as interesting compounds in context of the anti-COV activity remains not fully clear. First of all – it suggested that the library was screened without providing any details on the source of library and how the high-throughput screening was performed. This needs clarification, and, perhaps adding some details in the discussion and experimental parts.

3) 435: “Moreover, most phenothiazines have been approved in clinical application, suggesting that phenothiazines are safe and there is no need to re-evaluate their toxicities and side effects”. I am sceptic aboout that. In my opinion this is disputable if phenothiazine drugs themselves will be promising for repurposing towards treatment of COV-infections. Even if they are non-cytotoxic, they still may be problematic because of potentially severe neurological, cardiac and psychiatric activities. Further, they seem not that specific as they were also reported to bind numerous other molecular targets outside the CNS.

Author Response

In their manuscript, the Authors report on antiviral activities of Phenothiazines against SARS-CoV-2. This is a sound piece of work described in appropriate manner. I recommend publication of this material, but have a few comments:

  • What is CCK-8? It should be mentioned in the text

Response: We apologize for not providing the detailed information of CCK-8. We have added the detailed information about CCK-8 as below: “After 48 h of post-incubation, about 10 μL of Cell Counting kit 8 (WST-8/CCK‐8) solution which utilizes a water-soluble tetrazolium salt to quantify the number of live cells by producing an orange formazan dye upon bio-reduction was added to each well and incubated at 37℃ for another 4 h” in the Materials and Methods section.

  • Line 186: “To discover broad-spectrum antiviral drugs for SARS-CoV-2, we firstly tested a panel of US FDA-approved drugs to identify potential entry inhibitors using SARS-CoV-2 Omi cron BA.4 subvariant pseudovirus (PsV).” The exact way in which phenothiazines were first identified as interesting compounds in context of the anti-COV activity remains not fully clear. First of all – it suggested that the library was screened without providing any details on the source of library and how the high-throughput screening was performed. This needs clarification, and, perhaps adding some details in the discussion and experimental parts.

Response: We sincerely apologize for not providing the detailed information of the FDA-approved drug library. We have added more detailed information in the revised manuscript. Briefly, to produce pseudovirions, psPAX2 plasmids and pHIV-Luciferase plasmids were co-transfected into HEK293T cells with pcDNA3.1-SARS-CoV-2 S-expressing plasmids and pCMV-VSV G-expressing plasmids respectively. To discover antiviral drugs, the candidate drugs were mixed with an equal volume of SARS-CoV-2 Omicron BA.4 PsV and incubated at 37℃ for 30 min. Then, the mixture was transferred to the HEK293T-hACE2 cells and incubated for 48 h. The cells were lysed with lysis buffer and the luciferase activity was assessed utilizing Luciferase Assay Kits. To exclude the drugs that only inhibited early events of the HIV-1 life cycle and to identify SARS-CoV-2-S-specific drugs, HIV-luc / VSV-G pseudotyped viruses bearing vesicular stomatitis virus (VSV) glycoproteins were used for secondary screening of the initial screening compounds. Additionally, we replenished a schematic diagram of the screening experiments using the library of FDA-approved drugs (Topscience; Catalog No. L4200) in newly added Fig. 1A in our revised manuscript to allow readers to more easily comprehend the screening flowchart.

  • 435: “Moreover, most phenothiazines have been approved in clinical application, suggesting that phenothiazines are safe and there is no need to re-evaluate their toxicities and side effects”. I am sceptic aboout that. In my opinion this is disputable if phenothiazine drugs themselves will be promising for repurposing towards treatment of COV-infections. Even if they are non-cytotoxic, they still may be problematic because of potentially severe neurological, cardiac and psychiatric activities. Further, they seem not that specific as they were also reported to bind numerous other molecular targets outside the CNS.

Response: We sincerely apologize for the overstated conclusion. In our revised manuscript, we have corrected it as “Moreover, most phenothiazines have been approved in clinical application, suggesting that phenothiazines may be safe to treat SARS-CoV-2 infection as well, and hence, may cost less to re-evaluate their toxicities and side effects”.

Round 2

Reviewer 1 Report

The manuscript revision improved the description of the methods and included new experiments to determine the in vivo significance of the drugs in inhibiting PsV infection.  Nonetheless, the data do not clearly support drug inhibition of S cleavage as the main mechanism of drug action, and alternative processes should be discussed. 

Specific comments:

1.- The abstract should mention the weak binding of the drug to the S as well as the partial inhibition of S cleavage by CTSL.

2.- As I mentioned in the previous revision, Figure 4F shows weak phenothiazine inhibition of CTSL-mediated S-cleavage.  In addition, the binding affinities of the drugs to S did not correlate with their efficacy on inhibiting S proteolysis:  Perphenazine or alimemazine had low affinity but high inhibition compared to other drugs.  In the left track of the experiment in Figure 4F, the S and CTSL should be labelled as + and -, respectively. 

In the new Figure 4G, the amount of full-length S is higher in the samples with drugs, but the amount of S2 is also higher, indicating more S processing into S1 and S2.  In addition, the amounts of unprocessed S were very low.

Author Response

Response to Reviewer 1 Comments

The manuscript revision improved the description of the methods and included new experiments to determine the in vivo significance of the drugs in inhibiting PsV infection.  Nonetheless, the data do not clearly support drug inhibition of S cleavage as the main mechanism of drug action, and alternative processes should be discussed. 

 Response: We thank the reviewer’s suggestion. We also agree with the reviewer that phenothiazine-mediated inhibition of S cleavage may be only one of the potential drug action mechanisms, as these drugs only showed partial inhibition of S cleavage by CTSL. While phenothiazines-mediated inhibition of SARS-CoV-2 pseudovirus infection is significant. We also do believe that phenothiazines may hijack alternative processes to prevent viral infection. Phenothiazines may not just prevent the cleavage of CTSL, they may target other proteases such as TMPRSS2 and furin. Since in patients with SARS-CoV-2 infection, the most susceptible organ is the lung, which contains many kinds of protease that may facilitate the cleavage-mediated S activation. Thus, further research is warranted to evaluate more detailed mechanisms of phenothiazines to inhibit the entry of SARS-CoV-2 besides inhibiting CTSL cleavage. Theoretically, phenothiazines as a class of compounds seem not that specific as they were also reported to bind numerous other molecular targets such as SARS-CoV-2 Mpro (Katrina L. Forrestall et al., Canadian Journal of Chemistry, 2021) and neuropilin-1 (Hashizume Mei et al., Antiviral Research, 2023, PMID: 36481388). Additionally, few studies reported that the potential antiviral effects of phenothiazines may be associated with their ability to inhibit viral binding to plasma membrane receptors (Rodrigo Machado-Vieira et al., Brazilian Journal of Psychiatry, 2021, PMID: 33440401; M. Plaze et al., Encephale, 2020, PMID: 32425222; Jiayu Lu et al., Life sciences, 2021, PMID: 33310043). Therefore, the inconsistency of the binding affinities and the efficacy of inhibiting the S proteolysis of Perphenazine and Alimemazine hemitartrate may also involve the competitive interaction with other molecular targets. Nevertheless, our study provides a unique insight into phenothiazine derivatives as promising therapeutic drugs for the treatment of SARS-CoV-2 infection by targeting the cleavage of S protein. In accordance with the review’s suggestion, we have discussed this issue in the Discussion of our revised manuscript.

Specific comments:

1.- The abstract should mention the weak binding of the drug to the S as well as the partial inhibition of S cleavage by CTSL.

Response: We thank the reviewer for these constructive suggestions. We do apologize for the overstatement within the abstract of our original manuscript. We agree with the reviewer that the binding affinity of phenothiazines to S protein is modest. Based on the cleavage inhibition assays, phenothiazines may only partially inhibit the cleavage of S. Thus, in our revised manuscript, to avoid being overstated, we have carefully revised these descriptions within the abstract as below “Mechanistic studies suggested that phenothiazines predominantly inhibited SARS-CoV-2 pseudovirus (PsV) infection at the early stage and potentially bound to the spike (S) protein of SARS-CoV-2, which may prevent the proteolytic cleavage of the S protein”.

2.- As I mentioned in the previous revision, Figure 4F shows weak phenothiazine inhibition of CTSL-mediated S-cleavage.  In addition, the binding affinities of the drugs to S did not correlate with their efficacy on inhibiting S proteolysis:  Perphenazine or alimemazine had low affinity but high inhibition compared to other drugs.  In the left track of the experiment in Figure 4F, the S and CTSL should be labelled as + and -, respectively. 

Response: We thank the reviewer for these constructive suggestions. We hold a positive viewpoint regarding to the reviewer's opinion that phenothiazines do not completely recover the CTSL-mediated S protein cleavage levels and we have revised our conclusion as “phenothiazines may not just prevent the cleavage of CTSL, they may also target other proteases such as TMPRSS2 and furin. Since in patients with SARS-CoV-2 infection, the most susceptible organ is the lung, which contains many kinds of protease that may facilitate the cleavage-mediated S activation. Thus, further research is warranted to evaluate more detailed mechanisms of phenothiazines to inhibit the entry of SARS-CoV-2 besides inhibiting CTSL cleavage” in our revised manuscript.

Theoretically, phenothiazines as a class of compounds seem not that specific as they were also reported to bind numerous other molecular targets such as SARS-CoV-2 Mpro (Katrina L. Forrestall et al., Canadian Journal of Chemistry, 2021) and neuropilin-1 (Hashizume Mei et al., Antiviral Research, 2023, PMID: 36481388). Additionally, few studies reported that the potential antiviral effects of phenothiazines may be associated with their ability to inhibit viral binding to plasma membrane receptors (Rodrigo Machado-Vieira et al., Brazilian Journal of Psychiatry, 2021, PMID: 33440401; M. Plaze et al., Encephale, 2020, PMID: 32425222; Jiayu Lu et al., Life sciences, 2021, PMID: 33310043). Therefore, the inconsistency of the binding affinities and the efficacy of inhibiting S proteolysis of Perphenazine and Alimemazine hemitartrate may also involve the competitive interaction with other molecular targets. We have replenished the related description in the Discussion in our revised manuscript.

We sincerely apologize for the wrong label of the left track of Figure 4F. We have corrected it in our revised manuscript.

In the new Figure 4G, the amount of full-length S is higher in the samples with drugs, but the amount of S2 is also higher, indicating more S processing into S1 and S2.  In addition, the amounts of unprocessed S were very low.

Response: We sincerely thank the reviewer for pointing out this conflicting result. After carefully checking our original data (Appendix 1) and the instruction of anti-spike antibody (Sino Biological, Catalog Number: 40591-T62) used in Figure 4G, we recognized that the antibody could only specifically target the S1 fragment of S but without the S2 fragment, whereas we unintentionally marked the signals of 65 kDa as the S2 protein (the molecular weight of the S2 protein is approximately 80 kDa). We think the signals of 65 kDa are non-specific and probably caused by drug treatment or the degraded S1 fragments within lysosomes. The description of 40591-T62 antibody specificity from its instruction was quoted as saying “40591-T62 was produced in rabbits immunized with purified, recombinant SARS-CoV-2 (2019-nCoV) Spike S1-His Recombinant Protein and the specificity was validated by Spike RBD-mFc Protein, Spike S1-His Protein and Spike RBD-His Protein”. The instruction of 40591-T62 was attached (Appendix 2). We sincerely apologize for this unintentional misinterpretation and deleted the S2 marker of Figure 4G in our revised manuscript.

The reviewer also concerned that more S proteins have been pre-processed into S1 and S2 upon drug treatment, which resulted in lower amounts of unprocessed S. While this phenomenon is normal as we produced pseudotyped SARS-CoV-1 S / HIV-1 viruses in HEK293T cells which contain high amounts of furin protease. Furin is able to cleave S when assembling viruses. The produced viruses already contain large amounts of processed S and non-covalently conjugated S1 and S2. Not like SARS-CoV-1 S, SARS-CoV-2 S protein contains a proprotein convertase motif at the S1/S2 boundary which can be cleaved by furin during viral packaging, not just during viral infection (Jian Shang et al., 2020, Proceedings of the National Academy of Sciences, PMID: 32376634). It also has been reported that the preactivation of S by furin-mediated cleavage can prime S for TMPRSS2 processing, as well as enhance SARS-CoV-2 infection (Guido Papa et al., 2021, PLOS Pathogens, PMID: 33493182; Thomas P Peacock et al., 2021, Nature Microbiology, PMID: 33907312). Thus, the high amounts of processed may be due to the pre-cleavage of S by furin, which results in lower amounts of unprocessed S upon viral production and infection.

Appendix 1. The original data of Figure 4G

Appendix 2. The instruction of anti-spike antibodySino Biological, Catalog Number: 40591-T62
